# Detecting macroevolutionary genotype–phenotype associations using error-corrected rates of protein convergence

**Kenji Fukushima** [1] ✉ **& David D. Pollock** [2,3]

On macroevolutionary timescales, extensive mutations and phylogenetic uncertainty mask the signals of genotype–phenotype associations underlying convergent evolution. To overcome this problem, we extended the widely used framework of non-synonymous to synonymous substitution rate ratios and developed the novel metric $\omega_C$, which measures the error-corrected convergence rate of protein evolution. While $\omega_C$ distinguishes natural selection from genetic noise and phylogenetic errors in simulation and real examples, its accuracy allows an exploratory genome-wide search of adaptive molecular convergence without phenotypic hypothesis or candidate genes. Using gene expression data, we explored over 20 million branch combinations in vertebrate genes and identified the joint convergence of expression patterns and protein sequences with amino acid substitutions in functionally important sites, providing hypotheses on undiscovered phenotypes. We further extended our method with a heuristic algorithm to detect highly repetitive convergence among computationally non-trivial higher-order phylogenetic combinations. Our approach allows bidirectional searches for genotype–phenotype associations, even in lineages that diverged for hundreds of millions of years.

A central aim of modern biology is to differentiate the huge amount of non-functional genetic noise from phenotypically important changes. Evolutionary processes at the molecular level are largely neutral and stochastic, but natural selection can constrain evolutionary pathways available to the organism. If similar environmental conditions recur in divergent lineages, the adaptive response may also be similar, leading to convergence, the repeated emergence of similar features in distantly related organisms[1]. The prevalence of phenotypic convergence is demonstrated by various examples throughout the tree of life, such as the camera eyes of vertebrates and cephalopods, powered flight of birds and bats and trap leaves of distantly related carnivorous plants. Because the repeated emergence of such complex traits by neutral evolution alone is extremely unlikely, convergence at the phenotypic level is considered strong evidence for natural selection.

Phenotypic convergence is necessarily caused by molecular events and often coincides with detectably excess levels of convergent molecular changes in gene regulation, gene sequences, gene repertoires and other hierarchies of biological organization[2,3]. A meta-analysis reported that 111 out of 1,008 loci had been convergently modified to attain common phenotypic innovations, sometimes even between different phyla[4], illustrating that genotype–phenotype associations are frequently observed on macroevolutionary scales. For example, several lineages of mammals, reptiles, amphibians and insects acquired resistance to toxic cardiac glycosides using largely overlapping sets of

[1]Institute for Molecular Plant Physiology and Biophysics, University of Würzburg, Würzburg, Germany. [2]Department of Biochemistry and Molecular Genetics, University of Colorado School of Medicine, Aurora, CO, USA. [3]Biological Sciences Division, Earth and Biological Sciences Directorate, Pacific Northwest National Laboratory, Richland, WA, USA. ✉e-mail: kenji.fukushima@uni-wuerzburg.de

amino acid substitutions in a sodium pump[5]. Another example illustrated how human cancer cells and plants employed common amino acid substitutions in topoisomerase I to cope with a common toxic cellular environment generated by plant-derived anticancer drugs[6].

Genome sequences are becoming more available for diverse lineages from the entire tree of life[7], making it possible to explore macroevolutionary genotype–phenotype associations on large scales. However, because many molecular changes are nearly neutral (that is, almost no effect on fitness) and essentially non-functional in nature[8], false positive convergence in the form of stochastic, non-adaptive, convergent events is particularly problematic when conducting a genome-scale search. Furthermore, false positives can arise from methodological biases. For molecular convergence, a major source of bias occurs because such inference is sensitive to the topology of the phylogenetic tree on which substitution events are placed[9] (Fig. 1a), while alternative methods that do not place substitutions on phylogenetic trees suffer even more severe rates of false positives[10–12] (Supplementary Text 1). A correctly inferred tree avoids false positives due to phylogeny[13], but topological misinference due to technical errors, insufficient data or biological factors such as introgression, horizontal gene transfer (HGT), paralogy, incomplete lineage sorting and within-locus recombination can all create substantial amounts of false convergence signals even when adaptive convergence did not actually occur[2,9,14,15]. Importantly, false convergence events driven by topological errors tend to similarly affect both non-synonymous and synonymous substitutions (Supplementary Fig. 1a). By contrast, truly adaptive convergence should occur almost exclusively in non-synonymous substitutions (amino acid-changing substitutions), as positive selection on synonymous substitutions is negligible or at least not prevalent[16] (Supplementary Fig. 1b). Therefore, synonymous convergence can potentially serve as a reliable reference for measuring the rate of expected non-synonymous convergence due to phylogenetic inference error.

A widely used framework for understanding how functionally constrained proteins evolve compared with completely unconstrained expectations is to contrast rates of non-synonymous and synonymous substitutions. The ratio of these rates within a protein-coding sequence accounts for mutation biases and is often denoted as $\omega$, $dN/dS$ or $K_a/K_s$ (ref. [16]). Here we extend this framework to derive the new metric ratio $\omega_c$ and implement it to measure phylogenetic error-corrected rates of convergence. Simulation and empirical data analysis show that this new metric has high sensitivity while suppressing false positives. We further show its capability to detect factors that affect protein convergence rates and to identify likely adaptive protein evolution in a genome-scale dataset by an exploratory analysis without a pre-existing hypothesis. We also develop a heuristic algorithm to explore convergent signals with high signal-to-noise ratios in exponentially increasing numbers of higher-order phylogenetic combinations.

## Results

### Extending the substitution rate ratio framework to convergence

One of the most commonly accepted measures of the rate of protein evolution compared with completely unconstrained expectations is the ratio between non-synonymous and synonymous substitution rates, denoted as $dN$ and $dS$, respectively[16]. In a model-based framework, the ratio $dN/dS$ is parameterized as $\omega$.

Inspired by $\omega$, we developed a similar metric, $\omega_c$, that applies to substitutions that occurred repeatedly on a combination of separate phylogenetic branches (combinatorial substitutions; Supplementary Fig. 1c and Supplementary Text 2). The metric $\omega_c$ estimates the relative rates of convergence obtained by contrasting the rates of non-synonymous and synonymous convergence ($dN_c$ and $dS_c$, respectively). Using this ratio, important biological fluctuations, such as among-site rate heterogeneity and codon equilibrium frequencies,

are taken into account (Supplementary Text 3 and Methods). Note that in neutrally evolving genes, the theoretical expectation of $\omega_c$ is 1.0, even if $\omega$ in the underlying codon substitution matrix is not 1.0 (usually lower). Similar to previously proposed convergence metrics[13,17,18], $\omega_c$ is calculated from substitutions at multiple codon sites across protein-coding sequences. As a result, one $\omega_c$ value is obtained for each gene for each branch pair (or for a combination of more than two branches) in the phylogenetic tree. A unique feature of $\omega_c$ setting it apart from other metrics is its error tolerance. For example, if one of the branches in a branch combination is in error, $\omega_c$ is a measure of the ratio of false convergence events of both kinds falsely attributed to a non-existent branch combination. In this way, the $\omega_c$ values remain close to the neutral expectation of 1.0, even when topology errors are involved. Our method is implemented in the Python programme CSUBST (https://github.com/kfuku52/csubst), which takes as input a rooted phylogenetic tree and a codon sequence alignment (Fig. 1b and Supplementary Fig. 2).

### The robustness of $\omega_c$ in simulated molecular evolution

Conventionally, observed levels of convergent amino acid substitutions have been contrasted either to the amount of convergence expected under a substitution model (e.g., the convergence measure $R$[18]) or to other combinations of amino acid substitution patterns that are similarly affected by site-specific constraint (that is, double divergence; $C/D$ (refs. [13,17])) (Supplementary Table 1 and Supplementary Text 4). Here we focus on whether $\omega_c$ performs better as a measure of convergence between branches in comparison to alternative metrics. Accordingly, we generated simulated sequences with 500 codons along a balanced phylogenetic tree ending with 32 sequences at the tips (or leaves), in all cases comparing two deeply separated tip lineages (shown as dots in Fig. 1c; Supplementary Table 2). In this analysis, we compared $C/D$, $dN_c$, $dS_c$ and $\omega_c$ under four evolutionary scenarios of relationships between the two tips being compared: (1) full neutral evolution along all branches (Neutral); (2) neutral evolution for nearly all branches but with convergent selection along the two deeply separated tip lineages (Convergent); (3) neutral evolution with phylogenetic tree topology error in the form of a copy-and-paste transfer from one of the two deeply separated lineages to the other, overwriting its genetic information (Transfer) or (4) neutral evolution but using a randomly reconstructed phylogenetic tree to detect convergence (Random). The metric $dN_c$ is obtained by dividing the observed value of non-synonymous convergence ($O_c^N$) by the expected value ($E_c^N$) and is essentially equivalent to the previously proposed metric called $R$[18], but we use the $dN_c$ notation here to clarify its relationship to $dS_c$, the ratio of observed to expected values of synonymous convergence ($O_c^S / E_c^S$).

During neutral evolution, sequences evolved under a constant codon substitution model without any adaptive convergence or constraint on amino acid substitutions other than those imposed by the structure of the genetic code and relative codon frequencies. In the Neutral scenario (Fig. 1c), the trees used for simulation and reconstruction were identical. $C/D$ was much lower than 1.0, as expected, while the other three metrics ($dN_c$, $dS_c$ and $\omega_c$) were close to but lower than the theoretical expectation of 1.0 (Fig. 1d). This observation is probably due to the fact that the convergent events must be inferred and are not actually observed, as investigated previously for the convergence measure $R$[18]. In the Convergent scenario, adaptive convergence on the focal pair of deeply separated branches (red branches in Fig. 1c) was mimicked by convergently evolving 5% of codon sites (25 sites) in the two branches under substitution models biased towards codons encoding the same randomly selected amino acid. This generated an average of four excess non-synonymous convergent substitutions on these two branch pairs ($O_c^N$ in Fig. 1c). In the Convergent scenario, the three protein convergence metrics, $C/D$, $dN_c$ and $\omega_c$, yielded values substantially higher than they did under the Neutral scenario, while the synonymous change measure $dS_c$ remained comfortably well

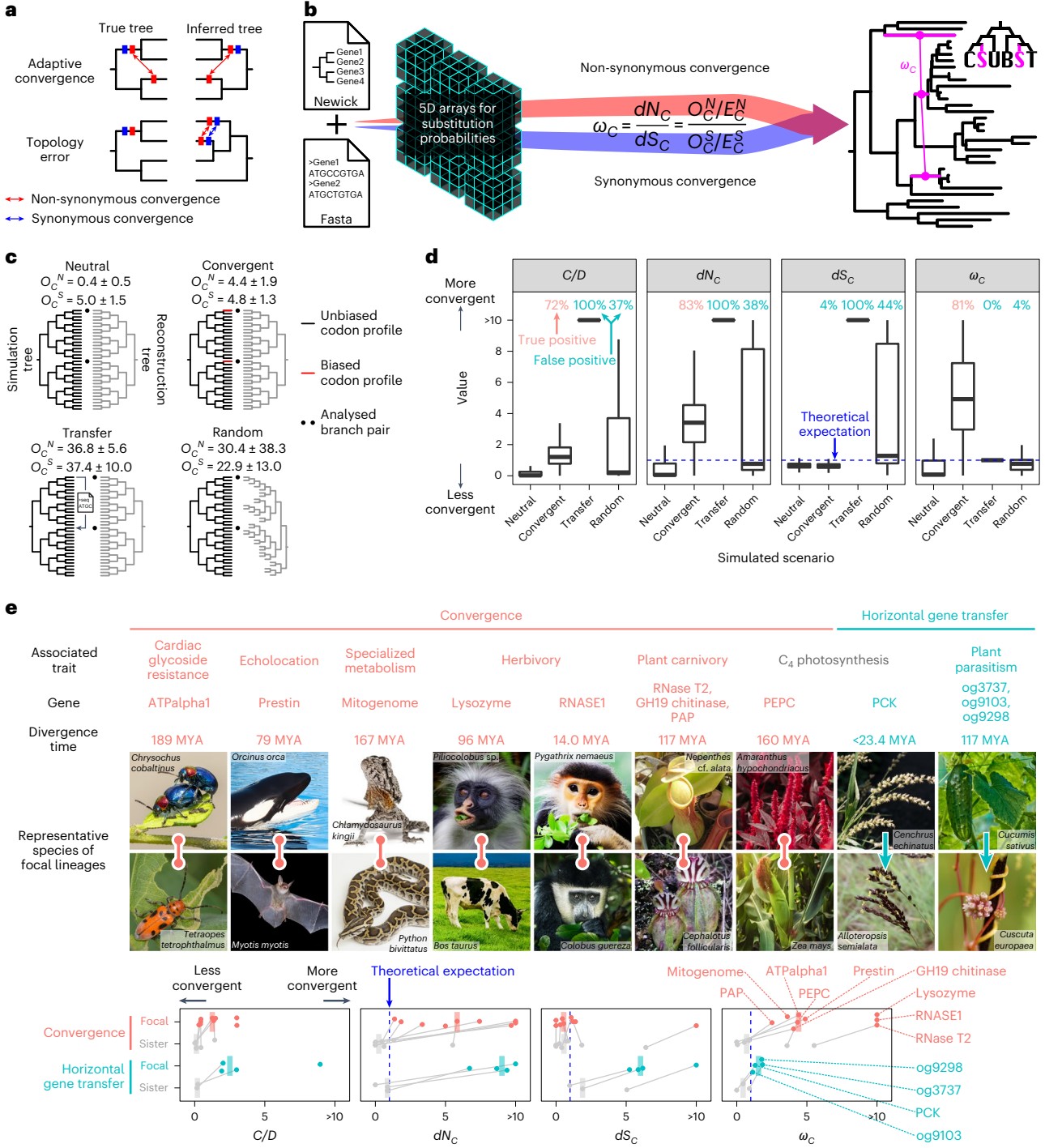

**Fig. 1 | Challenges and solutions for the detection of molecular convergence.**
**a**, False convergence is caused by tree topology errors. **b**, The overview of CSUBST. This programme processes substitution probabilities to derive observed ($O_C^N$ and $O_C^S$) and expected ($E_C^N$ and $E_C^S$) numbers of non-synonymous and synonymous convergence and evaluate their rates ($dN_C$ and $dS_C$) in branch combinations in a phylogenetic tree. **c**, Generation of simulated datasets for performance evaluation in different evolutionary scenarios. The numbers of observed non-synonymous and synonymous convergence are indicated above trees ($O_C^N$ and $O_C^S$, respectively; mean ± standard deviation). **d**, The estimated rates of protein convergence in different scenarios. Each box plot corresponds to the results of 1,000 simulations. Dashed lines indicate the theoretical expectation (=1.0) except for the ratio of convergence and divergence events (C/D) (refs. [13,17]) for which no theoretical expectation is available. Values greater than the 95th percentile in the Neutral scenario are defined as true and false positives in Convergent and other scenarios, respectively, and are indicated at the top of the plot in pink (true) and blue (false). Box plot elements are defined as follows: centre line, median; box limits, upper and

lower quartiles; whiskers, 1.5 × interquartile range. **e**, Performance of convergence metrics in empirical datasets. Known examples of protein convergences and HGTs are analysed with C/D, $dN_C$, $dS_C$ and $ω_C$. Median values (bars) are overlaid on individual data points that correspond to gene trees. In trees where convergence occurred in more than two lineages, the median of all focal branch pairs is reported. The branch pairs sister to the focal branches are shown as a control[10], except in cases where there is no substitution at all or the sister branches are phylogenetically not independent. Divergence time is according to timetree.org[95]. The comparison with the background levels for each dataset is shown in Supplementary Fig. 5. The characteristics of the datasets are summarized in Supplementary Table 3. MYA, million years ago. Image credits for panel **e**: *Cenchrus echinatus*, Chutima Chaimratana/Shutterstock; *Tetraopes tetrophthalmus*, Paul Reeves Photography/ Shutterstock; *Myotis myotis*, Agami Photo Agency/Shutterstock; *Colobus guereza*, Radek Karko/Shutterstock; *Alloteropsis semialata* Alexey Yakovlev under a Creative Commons license CC BY 4.0; *Cuscuta europaea*, ChWeiss/Shutterstock; alll other images except for *Nepenthes* cf. *alata* and *Cephalotus follicularis* from freepik.com.

below 1.0. Using the distribution of metric values under the Neutral scenario as a reference, we see that 70–80% of the detection metric values in the Convergent scenario are above the 95th percentile of the 1,000 simulations in their respective neutral distributions, while only 3.5% of $dS_C$ values are above this threshold, indicating that this level of convergence is usually detected by all three of the protein convergence metrics (Fig. 1d). In $\omega_C$, this level of detection was achievable with only two to three non-synonymous convergent substitutions, and the positive rates exceeded 95% with seven or more convergent substitutions (Supplementary Fig. 3). To be thorough, we considered that $\omega_C$ metrics can in general be derived for nine types of combinatorial substitution (that is, substitutions occurring at the same protein site in multiple independent branches; Supplementary Text 2) based on whether the ancestral and descendant states are the same or different, or in any state among multiple branches (Supplementary Fig. 1c). In the Convergent scenario, only the $\omega_C$ metrics involved in convergence (that is, not divergence) showed a response, confirming its specificity (Supplementary Fig. 4a).

We next considered Transfer and Random scenarios that include phylogenetic error. In the Transfer scenario, we transferred one of the focal tip sequences to the other focal tip sequence in the simulation, but the phylogenetic tree used in the analysis remained unchanged, as might happen with HGT events (Fig. 1c). In the Random scenario, we fully randomized the entire reconstructed tree relative to the true tree (Fig. 1c). Excess convergence detected in either of these scenarios is considered a false positive. We determined that both $C/D$ and $dN_C$ (and thus $R$ too) are sensitive to the errors (Fig. 1d). By contrast, and as intended, $\omega_C$ values were close to the neutral expectation because the rise in $dN_C$ due to phylogenetic error is matched by a similar increase of $dS_C$, and they cancel each other out in the $\omega_C$ metric (Fig. 1d). Further simulations supported the robustness of $\omega_C$ against the rate of protein evolution, model misspecification, tree size and protein size (Supplementary Fig. 4b–f). Still, care must be taken when using simple codon substitution models, such as MG and GY models (Supplementary Text 5). Furthermore, $\omega_C$ showed low false positive rates in sister branches that serve as a control for the focal branch pairs[10] (Supplementary Fig. 4g). Taken together, our simulation showed that $\omega_C$ effectively counteracts false positives caused by phylogenetic errors without loss of power.

### $\omega_C$ is robust to false convergence in empirical datasets
To test whether $\omega_C$ performs well with real data, we collected protein-coding sequence datasets from known molecular convergence events in various pairs of lineages covering insects, tetrapods and flowering plants (Fig. 1e, Supplementary Fig. 5, Supplementary Fig. 6 and Supplementary Table 3). Insects that feed on milkweed (Apocynaceae) harbour amino acid substitutions in a sodium pump subunit (ATPalpha1) that confer cardiac glycoside resistance[19–21] (Supplementary Fig. 5a). Echolocating bats and whales share amino acid substitutions in the hearing-related motor protein prestin to enable high-frequency hearing[22,23] (Supplementary Fig. 5b). An extensive molecular convergence occurred in the mitochondrial genomes of agamid lizards and snakes, presumably due to physiological adaptations for radical fluctuations in their aerobic metabolic rates[13] (Supplementary Fig. 5c–g). Phosphoenolpyruvate carboxylase (PEPC), a key enzyme for carbon fixation in $C_4$ photosynthesis, shares multiple amino acid convergence[28,29] (Supplementary Fig. 5h). In all these examples, $\omega_C$ successfully detected convergent lineages, while it was always lower and in many cases close to the neutral expectation in the branch pairs sister to the focal lineages, which serve as a negative control[10] (Fig. 1e and Supplementary Table 4). Moreover, the $\omega_C$ values of the focal branch pairs tended to be high compared with background levels in the phylogenetic trees (Supplementary Fig. 5i). Analysis of different categories

of combinatorial substitutions correctly recovered a trend consistent with the action of intramolecular epistasis, which did not appear in the simulations (Supplementary Text 6 and Supplementary Fig. 5j,k).

To test robustness against phylogenetic errors, we also employed reported cases of HGTs associated with $C_4$ photosynthesis[30] and plant parasitism[31]. We reconstructed the phylogenetic trees of the HGT genes with a constraint that enforces species tree-like topologies (Supplementary Fig. 7). This operation separates the HGT donor and acceptor lineages and creates false convergence (Supplementary Fig. 1a). Consistent with the simulation results, $\omega_C$ values in HGTs were lower than the adaptive convergence events (Fig. 1e). By contrast, $C/D$ and $dN_C$ showed values higher in HGTs than in the adaptive convergence events. Together with the simulations, these results show that the consideration of synonymous substitutions is essential for the accurate detection of molecular convergence in the presence of phylogenetic error and that $\omega_C$ outperforms current alternative methods.

### $\omega_C$ probes a high-confidence set of convergently evolved genes
Discovering adaptive molecular convergence in genome-scale datasets, which may be translated into genotype–phenotype associations, has been challenging since it is a rare phenomenon and false positives are high[10–12]. To examine whether the application of $\omega_C$ can generate plausible hypotheses of adaptive molecular convergence, we analysed the 21 vertebrate genomes covering a range from fish to humans (Fig. 2a and Supplementary Fig. 8a) and calculated $\omega_C$ and other metrics for all independent branch pairs in 16,724 orthogroups classified by OrthoFinder[32]. CSUBST completed the analysis even for the largest orthogroup (OG0000000) containing 682 genes encoding zinc finger proteins and 901,636 independent branch pairs (alignment length including gaps: 31,665 bp). We obtained a total of 20,150,538 branch pairs from all orthogroups, and subsequent analyses revealed that convergence probability decreased over time probably due to intramolecular epistasis (Fig. 2b, Supplementary Fig. 5l, Supplementary Fig. 8b,c and Supplementary Text 7), that gene duplication also reduced convergence probability (Fig. 2c and Supplementary Text 8) and that $\omega_C$ is robust for potential artifacts by falsely placed gene duplications and false gene grouping (Supplementary Fig. 8d,e and Supplementary Text 9).

We first extracted the branch pairs with the top 1% of $C/D$, $dN_C$ or $\omega_C$ values with a cut-off for a minimum of three non-synonymous and synonymous convergence ($O_C^N \geq 3.0$ and $O_C^S \geq 3.0$) (Fig. 3a). The top 1% threshold allows different convergence metrics to be compared without introducing arbitrary thresholds in each metric. The overlap between each set of branch pairs was moderate, with 1,348 branch pairs satisfying all three criteria out of 5,659 pairs with the top 1% $\omega_C$ values.

To examine which metrics better enrich for likely adaptive convergence, we compared the topological confidence scores of the selected branches. If artifacts due to tree topology errors are included, low confidence branches should be enriched. Analysis of the bootstrap-based confidence values[33,34] showed that $\omega_C$ selects branch pairs with higher confidence than the other two metrics (Fig. 3a). Furthermore, we examined the synonymous convergence rate ($dS_C$), which is not expected to be greater than the theoretical expectation in the adaptive convergence, and established that only $\omega_C$ satisfies such an assumption (Fig. 3a). These results indicate that $\omega_C$ has excellent properties for finding adaptive protein convergence in genome-scale analyses.

### Detecting convergent genes associated with a particular phenotype
As convergence metrics have been used to search for genes associated with phenotypes of interest, we next examined whether $\omega_C$ might be used to discover candidate genes underlying phenotypic convergence. Here we analysed a pair of herbivorous animal lineages as an example of a search for genes associated with a complex trait (that is, herbivory): ruminants (the stem branch of the clade including cattle (*Bos taurus*)

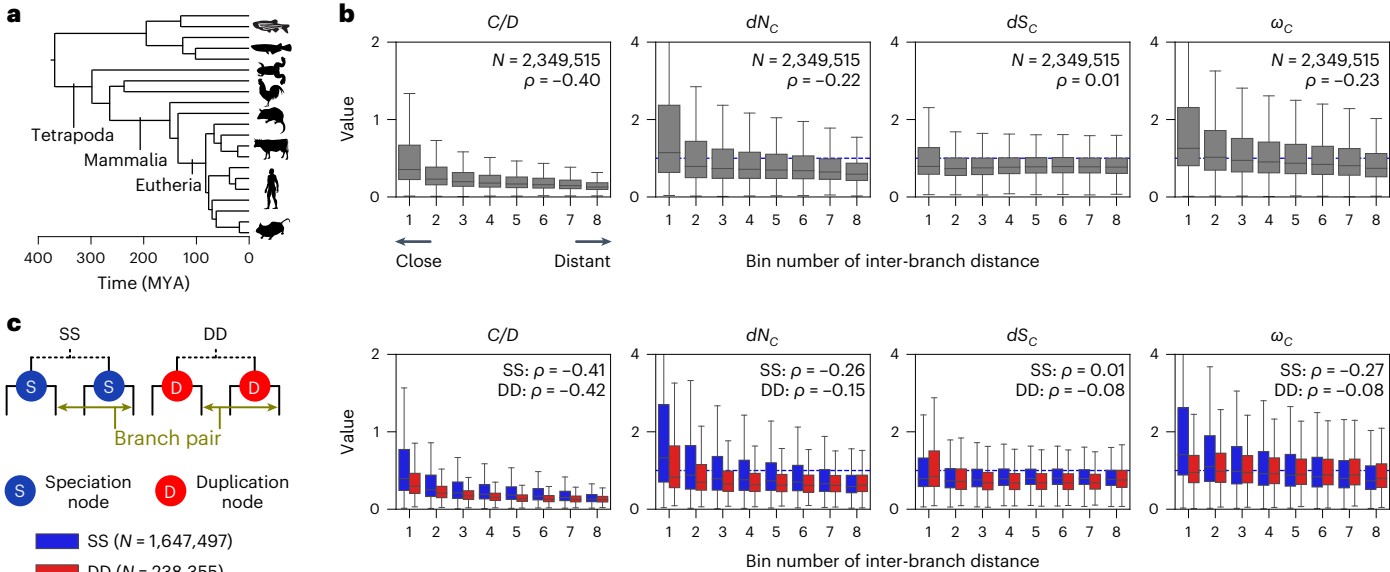

**Fig. 2 | Biological variation of $\omega C$ in a genome-scale dataset. a,** Phylogenetic relationships of the selected species. Supplementary Fig. 8a provides the complete phylogeny. The tree and divergence time estimates were obtained from timetree.org[95]. **b,** Temporal variation of convergence rates. The numbers of branch pairs ($N$) and Spearman's correlation coefficient ($\rho$) are shown. The bin range was determined to assign an equal number of branch pairs to each bin. To reduce the noise originating from branches where almost no substitutions occurred, branch pairs with both $O_C^N$ and $O_C^S$ greater than 1.0 were analysed (that

is, at least one convergent substitution each). Box plot elements are defined as follows: centre line, median; box limits, upper and lower quartiles; whiskers, 1.5 × interquartile range. **c,** Convergence rates depending on gene duplications. Branch pairs were categorized into speciation events (SS) and branch pairs after two independent gene duplications (DD) according to the presence of preceding gene duplications in no or both branches, respectively. Branch pairs with one preceding duplication were excluded from the analysis. Dashed lines indicate the theoretical expectation (= 1.0). Icon credits for panel **a**: PhyloPic.com.

and red sheep (*Ovis aries*)) versus rabbits (the terminal branch connected to *Oryctolagus cuniculus*). Using minimum thresholds for the number of convergent amino acid substitutions ($O_C^N \geq 3.0$) and protein convergence rate ($\omega_c \geq 3.0$), we obtained 352 candidate gene branch pairs corresponding to the above pair of lineages in a genome-scale analysis of the 21 vertebrates (Supplementary Table 5). By mapping the positions of substitutions onto known conformations of homologous proteins, we identified particularly compelling cases of likely adaptive convergence that generated genotypes linked to particular phenotypes (Supplementary Fig. 9). Examples included olfactory receptors in which convergent substitutions are located in the interior of the receptor barrel (Olfactory Receptor Family 7 Subfamily A (OR7A), Olfactory Receptor Family 2 Subfamily M Member 2 (OR2M2) and Olfactory Receptor Family 1 Subfamily B Member 1 (OR1B1), where substitutions may change ligand preference associated with herbivorous behaviour.

Similarly, the barrel-like structure of some solute carriers harboured convergent substitutions in their interior sides (Solute Carrier Family 5 Member 12 (SLC5A12), SLC51A, SLC22A and SLC44A1), suggesting their involvement in the uptake or transport of plant-derived compounds. Among these, SLC51A (also known as organic solute transporter α (OSTα)) may be a particularly attractive candidate. This protein plays a major role in bile acid absorption and, hence, in dietary lipid absorption[35]. The convergence in SLC51A may be coupled with another convergent event detected in CYP7A1, a cytochrome P450 protein known to serve as a critical regulatory enzyme of bile acid biosynthesis[36]. Cytochrome P450 Family 7 Subfamily A Member 1 (CYP7A1) harboured two convergent substitutions in its substrate-binding sites (Supplementary Fig. 9). While most herbivores secrete bile acids mainly in a glycine-conjugated form, ruminant bile is mostly in the form of taurine-conjugated bile acids, which remain soluble in highly acidic conditions[37]. The predominance of taurine-conjugated forms is also observed in rabbits, depending on species and developmental stage[38]. Thus, convergence in these proteins may be related to such nutritional physiology characteristic in these herbivorous animals.

Other examples of detected convergence included two convergent substitutions in the DNA-binding sites of a member of the zinc finger protein family, which functions as a transcriptional regulator[39] (Supplementary Fig. 9). Convergence in the substrate-binding sites of pancreatic elastase[40] and pancreatic DNase I (ref. [41]) may be related to their specialized digestion (Supplementary Fig. 9). In DNase I, amino acid sites exposed on the surface of protein structures displayed additional convergent substitutions that change the charge of their target amino acid residues (E124K, G172D and H208N), possibly resulting in convergent changes in the biochemical properties of the protein, such as optimal pH, resistance to proteolysis and post-translational modifications. Consistent with this idea, bovine and rabbit DNase I proteins are known to be more resistant to degradation by pepsin than their homologues in other animals[42]. Furthermore, E124K was shown to be important for the phosphorylation of bovine DNase I (ref. [43]). Other convergent substitutions will be promising candidates for future characterization. Taken together, these results show how our approach can detect genetic changes (for example, molecular convergence in SLC51A, CYP7A1 and DNase I) associated with phenotypes (for example, specialized digestion for herbivory) on the macroevolutionary scale.

### Exploratory analysis of molecular convergence

We further exploited the 21 vertebrate genomes to examine whether $\omega_c$ might be used to discover adaptive molecular convergence that may generate hypotheses of linked phenotypes. Because convergence at multiple levels of biological organization can provide strong evidence for adaptive evolution, we searched for simultaneous convergence in protein sequences and gene expression in an exploratory manner without a predefined hypothesis on convergently evolved genes and lineages. Using the same thresholds applied to the analysis of herbivores above ($O_C^N \geq 3.0$ and $\omega_c \geq 3.0$), we obtained 53,805 candidate branch pairs from all orthogroups.

Although this was an exploratory analysis in which all independent branch pairs were exhaustively analysed, many studies of convergent

evolution involve only a few groups of focal species. If such a research design is applied to this dataset (similar to the analysis of herbivores), the number of detected branch pairs will be much smaller. For example, because there are 861 branch pairs in the species tree, on average, 62.5 cases of protein convergence will be obtained in our genome-scale dataset for any particular analysis of two groups of species, although the numbers of analysable branch pairs and hence of detected convergence depend on several factors (Supplementary Text 10 and Supplementary Fig. 10).

To detect convergent gene expression evolution, we employed the amalgamated transcriptomes for six organs in the 21 vertebrate species[44]. Using this previously published dataset, we subjected curated gene expression levels (SVA-log-TMM-FPKM, $log_2$-transformed values of fragments per kilobase of exon per million reads mapped, corrected by trimmed mean of M-values and surrogate variable analysis) to multi-optima phylogenetic Ornstein–Uhlenbeck (OU) modelling, in which expression evolution is inferred as regime shifts of estimated optimal expression levels[45] (Fig. 3b). Phylogenetic positions and the numbers of expression regime shifts were determined by a LASSO (least absolute shrinkage and selection operator)-based algorithm with Akaike Information Criterion, which was also used for finding convergent shifts towards similar optimal values. In total, we detected 12,017 cases of expression convergence in 4,308 orthogroups. Setting the thresholds for gene expression specificity at $\tau \geq 0.67$ (ref. [46]) and expression levels at $\mu_{max} \geq 2.0$ (the maximum value of fitted SVA-log-TMM-FPKM)[44], we obtained a set of 2,917 high-confidence branch pairs for potentially adaptive convergence of expression patterns (Fig. 3c).

By taking the intersection of protein convergence and expression convergence, we discovered 33 cases of potentially adaptive joint convergence of expression patterns and protein sequences in 31 orthogroups (Fig. 3c and Supplementary Table 6). Gene duplication was frequently associated with joint convergence, with at least one branch experiencing gene duplication in 23 out of the 33 branch pairs ($P = 3.11 \times 10^{-25}$, $\chi^2 = 107.7$, $\chi^2$ test of independence). While gene duplication generally reduced the convergence rate, as discussed earlier (Fig. 2c), some of the independently generated duplicates may tend to evolve into the same sequence space when similar expression evolution takes place. Convergence of testis-specific genes was most frequently observed (19/33 orthogroups) and significantly enriched, with the effect size highest among six tissues (Supplementary Table 7). The mechanism by which the testis serves as a major place for functional evolution of duplicated genes has been explained by several factors, including the ease with which expression is acquired in spermatogenic cells[47,48]. This phenomenon is called the out-of-the-testis hypothesis, and our results suggest that predictable protein evolution may be enriched in this evolutionary pathway. While adaptive evolution may explain this evolutionary scenario, it is possible that partially relaxed constraints may also be involved in protein convergence, particularly at protein sites that were so constrained that almost no amino acid substitutions occurred before relaxation.

To infer the functional effect of convergent amino acid substitutions, we mapped the positions of substitutions onto known conformations of homologous proteins. Strikingly, we observed convergently evolved proteins where clusters of substitutions are localized to functionally important sites (Fig. 3d–g and Supplementary Text 11). For example, an orthogroup of dihydrodiol dehydrogenase (DHDH) showed joint convergence of expression and proteins (Fig. 3g). Possible physiological roles of this enzyme included the detoxification of cytotoxic dicarbonyl compounds, such as 3-deoxyglucosone derived from glycation[49,50]. Although the domain structure of proteins was well conserved among species (Supplementary Fig. 11a), the gene expression patterns of the encoding genes tended to vary. DHDH is known to show distinct tissue-specific expression patterns in mammals: kidney in monkeys (*Macaca mulatta*)[51], kidney and liver in dogs (*Canis lupus*)[52], liver and lens in rabbits[53] and various tissues in pigs (*Sus scrofa*)[49]. Our amalgamated transcriptomes showed largely consistent species-specific expression patterns (Fig. 3g). The OU analysis recovered four lineage-specific regime shifts categorized into two pairs of convergent expression evolution. One of them, the convergence of gene expression that occurred between frogs (*Xenopus*) and the blind cave fish (*Astyanax*), which diverged approximately 435 million years ago[54], is characterized by kidney-specific expression. The *Xenopus* gene ENSXETG00000033613 appeared to have arisen from a more widely expressed ancestral gene after a lineage-specific gene duplication. By contrast, the *Astyanax* gene ENSAMXG00000005808 may have acquired kidney-specific expression without any detectable duplication. In this branch pair, we detected a protein convergence rate that cannot be explained by neutral evolution alone, with a convergence of five amino acid sites (Supplementary Fig. 11a). These convergent substitutions localized around the active site, while we did not observe such a trend for the double divergence (Fig. 3g). This result suggests that the convergent substitutions may have occurred adaptively to change ancestral catalytic function.

DHDH has broad substrate specificity for carbonyl compounds. This protein oxidizes *trans*-cyclohexanediol, *trans*-dihydrodiols of aromatic hydrocarbons and monosaccharides including D-xylose, while it reduces dicarbonyl compounds, aldehydes and ketones[52]. Its active site is predominantly formed by hydrophobic residues, suggesting their role in catabolizing aromatic hydrocarbons[55,56]. Notably, the convergent substitutions in the substrate-binding pocket tended to increase amino acid hydrophobicity (Supplementary Fig. 11b), suggesting that the remodelling of the active site may have led to the acquisition of new substrates, and hence a novel detoxification ability, in *Xenopus* and *Astyanax*.

In summary, $\omega_C$ was not only robust against phylogenetic errors, outperforming other methods in simulation and empirical data, but also allowed us to discover plausible adaptive convergence from a genome-scale dataset without a pre-existing hypothesis. The genotypes detected by molecular convergence analysis provide opportunities for the phenotypic association, mechanistic assessment and

**Fig. 3 | Joint convergence of gene expression patterns and protein sequences. a**, Comparison of convergent branch pairs obtained by different methods in the vertebrate dataset. Branch pairs with $O_C^N \geq 3.0$ and $O_C^S \geq 3.0$ were analysed. The stochastic equality of the data was tested by a two-sided Brunner–Munzel test with $W$ as the test statistic[96]. Box plot elements are defined as follows: centre line, median; box limits, upper and lower quartiles; whiskers, 1.5 × interquartile range. **b**, A schematic illustration of convergent expression evolution modelled with a multi-optima Ornstein–Uhlenbeck process. **c**, Venn diagrams showing the extent of overlap between protein and expression convergence. Circles represent the sets of branch pairs. **d–g**, Examples of the likely adaptive joint convergence. Aldo-keto reductase family 1 (AKR1, **d**), Nudix hydrolase 16 like 1 (NUDT16L1, **e**), Myeloid associated differentiation marker (MYADM, **f**) and dihydrodiol dehydrogenase (DHDH, **g**) are shown. The silhouettes represent the species (Supplementary Fig. 8a) that carries the gene, and the clades involved in the joint convergence are indicated with an enlarged size. The colours of

branches and animal silhouettes indicate expression regimes. Branches involved in joint convergence are highlighted with thick lines, connected by the colour of the expression regime and annotated with convergence metrics. Localization of convergent and divergent substitutions on the protein structure is shown along with a close-up view of functionally important sites. Substrates and their analogues are shown as green sticks. Side chains forming the substrate-binding site are also shown as sticks. Note that these are the side chains in the protein from databases, so amino acid substitutions in the convergent lineages may result in distinct structures and arrangements. Site numbers correspond to those in the Protein Data Bank (PDB) entry or the AlphaFold structure (from **d** to **g**: 1Q13, 5W6X, AF-Q6DFR5-F1-model_v2 and 2O48). Icon credits for panels **d–g**: *Rattus norvegicus*, Rebecca Groom, under a Creative Commons license CC BY 3.0; *Astyanax mexicanus*, Milton Tan/PhyloPic under a Creative Commons license CC BY-NC-SA 3.0; all other images are from PhyloPic.com or were created by the author.

experimental validation in vivo. This holds even if internal branches are involved in a detected convergence event, because most species in the clade will tend to retain the convergent genotypes identified,

and most species in the clade will tend to retain the phenotypic change that drove the molecular convergence. Therefore, molecular convergence revealed by our exploratory analysis will provide a basis for

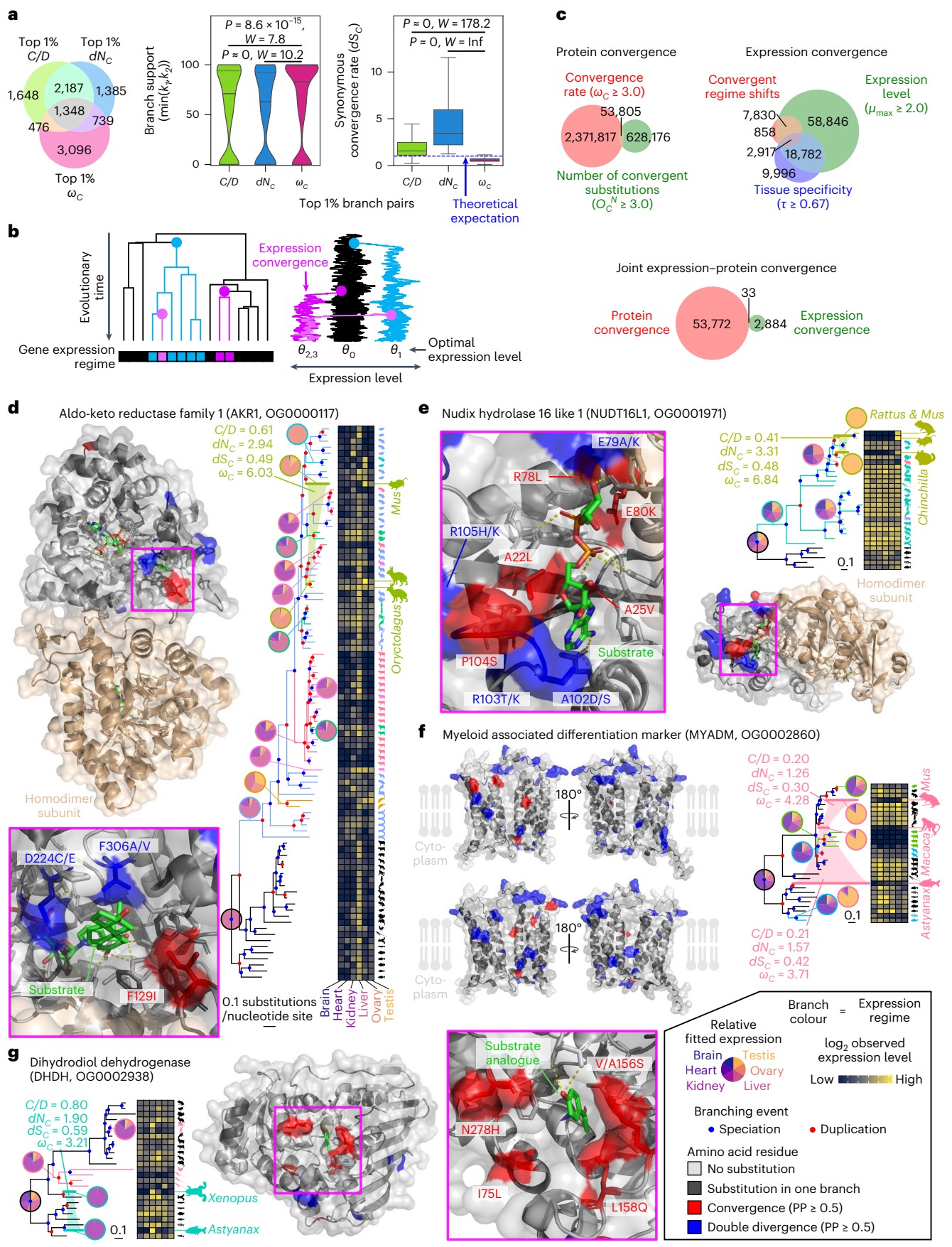

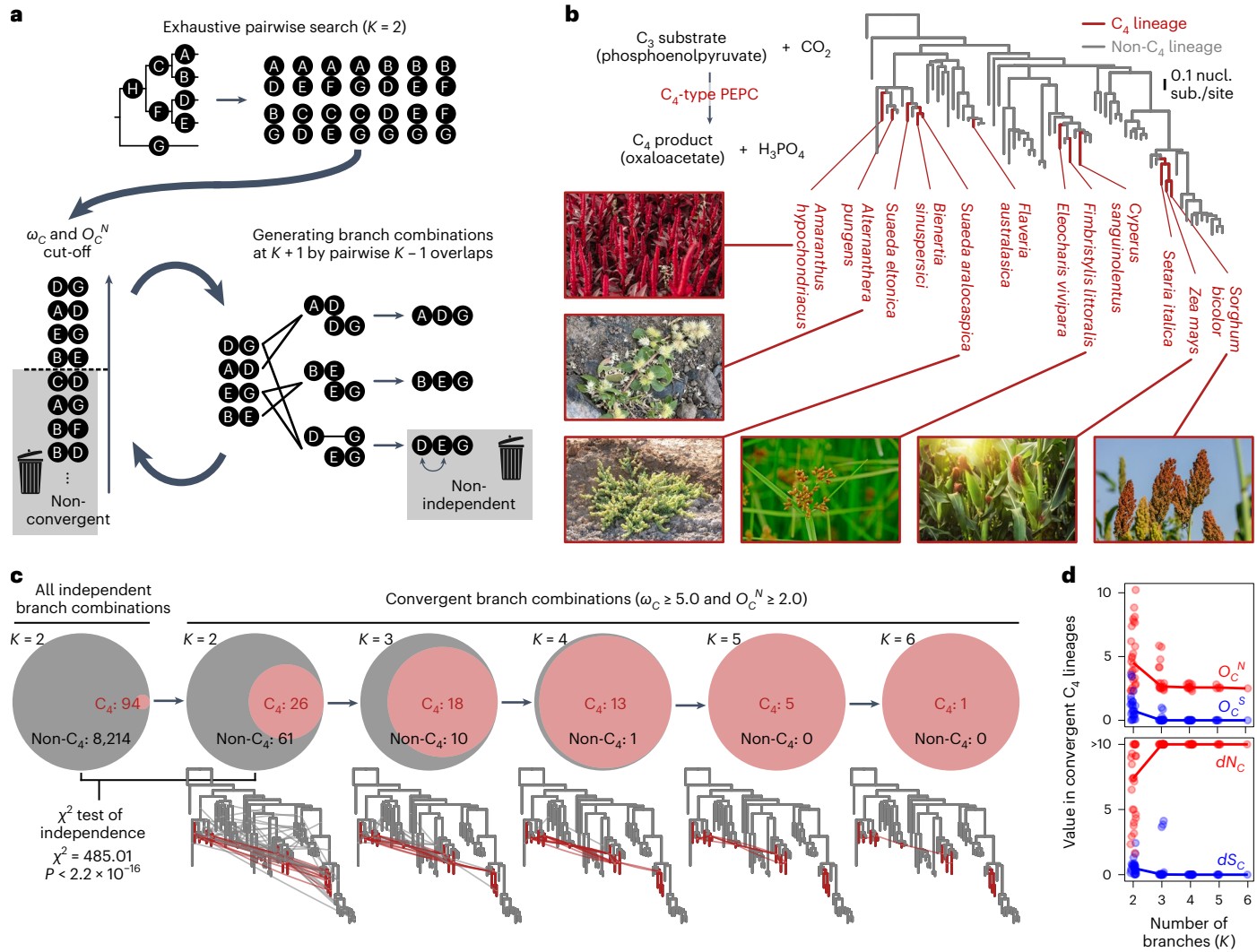

**Fig. 4 | Heuristic search of higher-order branch combinations for adaptive protein convergence. a**, Branch-and-bound algorithm for higher-order branch combinations. This method explores the higher-order combinatorial space until there are no more convergent branch combinations. **b**, The maximum-likelihood phylogenetic tree of PEPCs in flowering plants. The catalytic function of PEPC, which is crucial in $C_4$ photosynthesis, is illustrated. Photographs of representative $C_4$ photosynthetic lineages are shown. The bar indicates 0.1 nucleotide substitutions per nucleotide site. The complete tree is shown in Supplementary Fig. 6. **c**, Higher-order convergence enriches $C_4$-type PEPCs. The Venn diagrams show the proportion of convergent branch combinations of $C_4$-type and non-$C_4$-type lineages (red and grey, respectively). Branch combinations containing both were included in non-$C_4$. In the phylogenetic trees, convergent branch combinations are shown as edges connecting branches. **d**, Improvement of the signal-to-noise ratio in higher-order branch combinations. The line graph shows the median values of the total probabilities ($O_C^N$ and $O_C^S$) and the rates ($dN_C$ and $dS_C$) of non-synonymous and synonymous convergence in the convergent branch combinations of $C_4$ lineages. Points correspond to branch combinations. Image credits for panel **b**: *Suaeda aralocaspica* from ref. [97] and *Alternanthera pungens* by monkeyjodey under a Creative Commons lincense CC BY 4.0; *Fimbristylis littoralis*, Supratchai Pimpaeng/Shutterstock; all other images from freepik.com.

understanding overlooked phenotypes that are in common among species in clades descended from branches where the convergent events occurred.

## Heuristic detection of highly repetitive adaptive convergence

Convergent events observed on even more than two independent lineages are exceptionally good signals of adaptive evolution, if they exist, because three or more combined convergences should be extremely rare in random noise. Conventionally, convergence in more than two branches has been analysed as multiple pairwise comparisons for which there is a prior hypothesis of convergence. The difficulty in analysing higher-order combinatorial substitutions without specific prior hypotheses lies in the need to explore a vast combinatorial space that exponentially expands as the number of branches to be combined ($K$) increases. For example, an evenly branching tree with 64 tips has 7,359

independent branch pairs (that is, at $K = 2$), but the number of branch combinations exponentially increases to 333,375 and 6,976,859 in triple ($K = 3$) and quadruple ($K = 4$) combinations, respectively, making it impractical to exhaustively search highly repetitive convergence even in a single phylogenetic tree when a hypothesis on focal lineages is unavailable.

To overcome this limitation, we developed an efficient branch-and-bound algorithm[57] that progressively searches for higher-order branch combinations (Fig. 4a and Supplementary Fig. 12a). For the performance evaluation, we used the PEPC tree (Fig. 4b) because it has repeated adaptive convergence for its use in $C_4$ photosynthesis (Fig. 1e). While the exhaustive search required 156 minutes with $K = 3$ to analyse 307,432 branch combinations using two central processing units (CPUs), our branch-and-bound algorithm required only 21 seconds. At $K = 4$, the exhaustive search completed within a

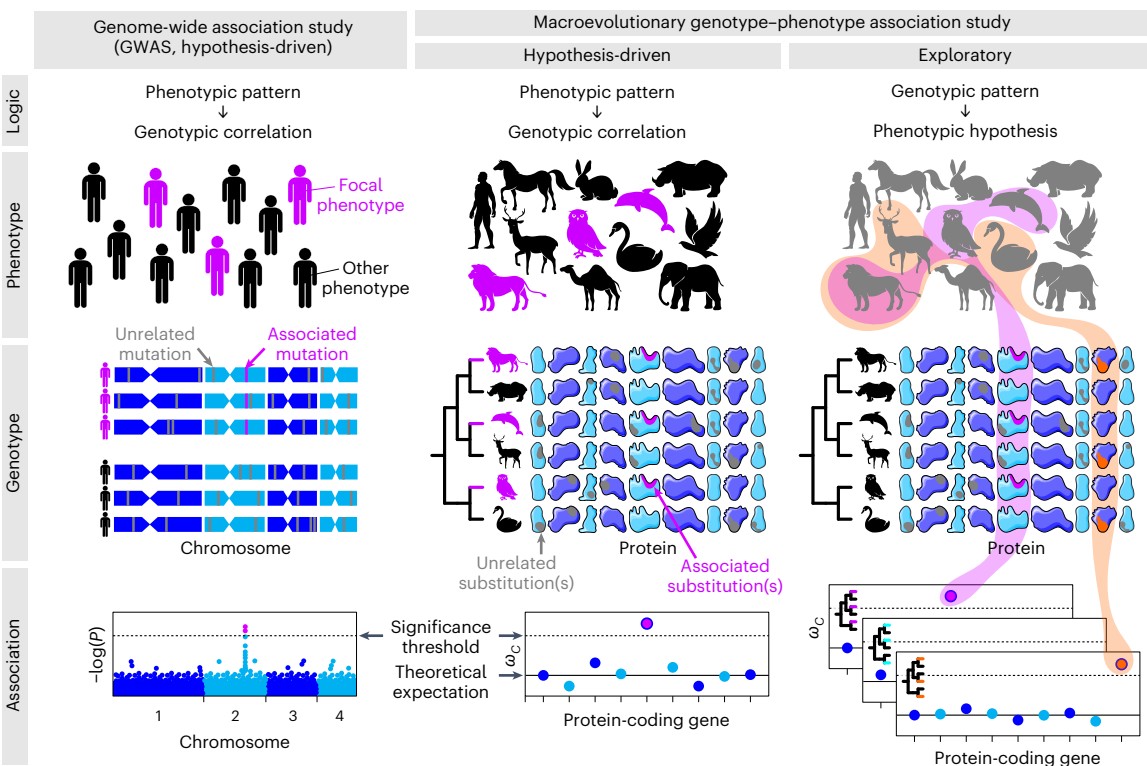

**Fig. 5 | Analysis of the genotype–phenotype association within and between species.** The proposed method improves the accuracy of the hypothesis-driven approach in the macroevolutionary scale and enables exploratory approaches. Note that for visualization purposes, the number of individuals and species shown here is smaller than the actual number required for analysis. Focal phenotypes and associated mutations are shown in pink and orange, other phenotypes are shown in black and unrelated mutations and substitutions are shown in grey. Icons from freepik.com. Credits for protein shapes: Smart Servier Medical Art under a Creative Commons license CC BY -SA 3.0.

practical time by using 16 CPUs (46 hours for nearly 8 million combinations) but failed to complete at K = 5 (152 million combinations). By sharp contrast, the heuristic search took about 5 minutes for the entire analysis, of which the higher-order analysis with $K$ ranging from 3 to 6 took only about 1 minute to analyse as few as 390 combinations with two CPUs (Supplementary Table 8).

The analysed tree covered nine independent origins of $C_4$-type PEPC, and the corresponding branch pairs of $C_4$ lineages accounted for 1.1% of all possible pairs (94/8,308). Convergent branch pairs defined by a threshold ($\omega_C \geq 5.0$ and $O_C^N \geq 2.0$) enriched for the $C_4$ lineages at $K = 2$ (29.9%, 26/87; Fig. 4c). The convergence of non-$C_4$ lineages (61/87, including pairs of $C_4$ and non-$C_4$ branches) can be interpreted as false positives or adaptive convergence associated with other currently unknown functions. The subsequent higher-order analysis resulted in the discovery of highly repetitive convergence in combinations of as many as six branches (that is, $K = 3$ to $K = 6$). As the order increased, the lineages of $C_4$-type PEPCs rapidly predominated and accounted for all the combinations detected at $K \geq 5$ (Fig. 4c), even though the heuristic algorithm was not given any information about the $C_4$ lineages.

In the higher-order $C_4$ branch combinations, the detected convergence events were almost entirely non-synonymous ($O_C^N$), while synonymous convergence ($O_C^S$) was negligible (Fig. 4d). As a result, the rate of synonymous convergence ($dS_C$) quickly approached zero (Fig. 4d). Notably, the higher-order convergent substitutions were located at functionally important protein sites. In the convergent branch combinations with $K = 6$, we identified three amino acid sites with a joint posterior probability of non-synonymous convergence greater than 0.5: V627I, H665N, and A780S (Supplementary Fig. 12b–d). The H665N substitution generates a putative $N$-glycosylation site that may be important for protein folding[29]. The A780S substitution, for which the signature of positive selection had been detected previously[28,58,59],

has been shown to change the enzyme kinetics related to the first committed step of $C_4$ carbon fixation[60–62] and is therefore considered a diagnostic substitution of $C_4$-type PEPC[28,29]. The third substitution, C627I, might be a good focus for future experimentation. Application of the heuristic search to the 21 animal genomes revealed that while likely adaptive higher-order convergence could be detected, false detections arising from inconsistently represented splicing variants should be carefully avoided when performing genome-scale analyses (Supplementary Text 12, Supplementary Fig. 13 and Supplementary Table 9). Nevertheless, these results demonstrate that higher-order analysis can substantially increase the signal-to-noise ratio in convergence analysis when there is repeated selective pressure to evolve similar biochemical functions.

## Discussion

In this study, we introduced a measure of convergent protein evolution, $\omega_C$, designed to account for false signals due to phylogenetic error. We showed, through simulation and analysis of real biological data, that $\omega_C$ mostly eliminates false positives without reduction in power to detect true signals. We also developed an approach to estimate the rates of highly repetitive convergence (that is, on more than two lineages) fully accounting for phylogenetic combinatorics and demonstrated that the specificity of $\omega_C$ increases further in the higher-order analysis. Because of its improved accuracy, $\omega_C$ should further drive macroevolutionary analyses where uncorrected measures have been used to identify responsible genotypes for particular phenotypes in a way similar to genome-wide association studies (GWASs). It is noteworthy that there are more direct extensions of the GWAS approaches to analyse among-species variations. Those methods, including PhyloGWAS[63], can be applied to closely related species to detect convergent selection on ancestral variation or through introgression[64]. Although those methods

are powerful, the applicability to distantly related species is limited. In our method, as in alleles identified by GWASs or the above-mentioned comparable approaches (or genes in gene-level association tests[65]), genes with excess convergence serve as clues to study macroevolutionary traits for which the molecular basis is unknown (Fig. 5). Furthermore, the accuracy of $\omega_c$ even allows exploratory analysis (Fig. 5), as demonstrated here in vertebrate genomes (Fig. 3). By conducting a genome-wide search of convergent branch combinations, we detected signatures of likely adaptive convergence, which leads to hypothesis generation on responsible phenotypes. This outcome was possible because $\omega_c$, unlike $P$-values from GWASs, does not require phenotypic traits as input. Convergently evolved genes identified by exploratory analysis will, in turn, lead to the discovery of overlooked phenotypes through future experimentation.

Although $\omega_c$ is a powerful means to detect convergence while removing the effect of phylogenetic error, there are other sources of stochastic error that can mask small signals. We successfully captured multiple known convergence events here, even with only two or three amino acid substitutions involved in small proteins (Fig. 1e and Supplementary Table 3). However, a convergent amino acid substitution at a single site in only two lineages may not reliably be identified as resulting from adaptation rather than random homoplasy, by $\omega_c$ or any other measure. Therefore, the number of observed non-synonymous convergence ($O_c^N$) should always be considered in addition to the phylogenetic error-corrected convergence rate ($\omega_c$), especially in a genome-scale screening with only two or three focal lineages. If many amino acid sites and/or many separate lineages are involved, true convergence is, in general, more easily detected. However, it should be noted that errors due to splicing variants may not be completely eliminated in higher-order branch combinations (Supplementary Text 12). In this study, we used different threshold values of $\omega_c$ and $O_c^N$, depending on the application, and our recommended usage is provided in Supplementary Text 13.

Although it is well established that phenotypes are associated with genotypes, the genetic basis for particular convergently evolved phenotypes may arise from distinct, non-convergent genetic changes[66,67]. These specific cases may sometimes occur because of convergent mechanisms, such as the use of similar but not identical amino acids, and the use of similar changes at adjacent residues in the protein structure[68]. The accumulation of knowledge about which mutations are repeatedly selected and which are not during convergent evolution may provide insight into the evolvability and constraints that govern the diversification of organisms.

While some evolutionary innovations may be unique, many traits arose convergently[69]. Fascinating examples not mentioned above include endothermy, hibernation, burrowing, diving, venom injection, electrogenic organs, eusociality, anhydrobiosis, bioluminescence, biomineralization, plant parasitism, mycoheterotrophy and multicellularity. In the past, the observation of similar phenotypes in multiple species led to the theory of evolution by natural selection[70]. The analysis of protein sequences in multiple species gave rise to the formulation of the nearly neutral theory of molecular evolution[71,72]. Likewise, cross-species genotype–phenotype associations illuminated through the analysis of molecular convergence, coupled with experimental evaluation of mutational effects (Supplementary Text 14), may lead to new conceptual frameworks on the constraint and adaptive changes at the molecular level that drive phenotypic change among species.

## Methods

### Simulated codon sequence evolution

With the input phylogenetic tree (Fig. 1c), codon sequences of specified length (500 codons) were generated with the 'simulate' function of CSUBST (https://github.com/kfuku52/csubst), which internally utilizes the Python package pyvolve for simulated sequence evolution[73]. An empirical codon substitution model with multiple nucleotide substitutions[74] was adjusted with observed codon frequencies (ECMK07+F) in the vertebrate genes encoding phosphoglycerol kinases (available from the 'dataset' function of CSUBST). The conventional $\omega$ ($dN/dS$) was set to 0.2. In the Convergent scenario, 5% of codon sites were evolved convergently in focal lineages (the pair of terminal branches in Fig. 1c). At convergent codon sites, the frequency of non-synonymous substitutions to codons encoding a single randomly selected amino acid was increased so that non-synonymous substitutions to the selected codons accounted for approximately 90% of the total. This operation increases the probability of amino acid convergence without changing relative frequencies among synonymous codons. The site-specific substitution rate at convergent codon sites was also doubled (that is, $r_t = 2$), and a higher non-synonymous/synonymous substitution rate ratio was applied (that is, $\omega = 5$) to mimic adaptive evolution. The simulation parameters for the other scenarios are summarized in Supplementary Table 2. For the Random scenario, randomized trees were generated in 1,000 simulations with the 'shuffle' function of NWKIT v0.10.0 and the --label option (https://github.com/kfuku52/nwkit).

### In-frame codon sequence alignment

Retrieved coding sequences (Supplementary Methods) were formatted into in-frame sequences using the 'pad' function of CDSKIT v0.9.1 (https://github.com/kfuku52/cdskit). Stop codons and ambiguous codons were replaced with gaps with the 'mask' function of CDSKIT. Amino acid sequences from translated coding sequences were aligned using MAFFT v7.455 with the --auto option[75], trimmed with ClipKIT v0.1.2 with default parameters and reverse-translated with the 'backtrim' function of CDSKIT. Gappy codon sites were excluded with the 'hammer' function of CDSKIT.

### Phylogenetic tree reconstruction

The gene tree was first reconstructed using IQ-TREE v2.0.3 with the general time-reversible nucleotide substitution model and four gamma categories of among-site rate variation (ASRV). To suppress branch attraction in the trees containing HGTs, topological constraints consistent with species classification were generated from the National Center for Biotechnology Information (NCBI) Taxonomy[76] using the 'constrain' function of NWKIT and used for tree search. Ultrafast bootstrapping with 1,000 replicates was performed to evaluate the credibility of tree topology[34] with further optimization of each bootstrapping tree (-bnni option)[33]. To improve tree topology, some datasets were subjected to phylogeny reconciliation with the species tree using GeneRax v1.2.2 (ref. [77]) (Supplementary Table 3). Branching events in gene trees were categorized into speciation or gene duplication by a species-overlap method[78] (Supplementary Methods). *Arabidopsis thaliana* orthologues in each clade were inferred from the tree topology. Minor differences in the methods applied to each dataset, from sequence retrieval to phylogenetic analysis, are summarized in Supplementary Table 3.

### Detecting convergent expression evolution

Using the dated species tree and rooted gene trees as inputs, the divergence time of individual gene trees was estimated by RADTE (https://github.com/kfuku52/RADTE) as described previously[44]. Evolution of gene expression levels (SVA-log-TMM-FPKM)[44] in brain, heart, kidney, liver, ovary and testis samples was modelled on the dated gene tree with phylogenetic multi-optima Ornstein–Uhlenbeck models (that is, Hansen models[79]) with the 'estimate_shift_configuration' function in the R package *l*1ou v1.40 (ref. [45]) as described previously[44]. Convergent regime shifts were then detected as multiple regime shifts that lead to similar expression levels, as judged by the 'estimate_convergent_regimes' function[45].

### Classification of combinatorial substitutions

Combinatorial substitutions were collectively defined as substitutions at the same protein site that occur in multiple independent branches in

a phylogenetic tree. When this occurs only in two branches, it is called a paired substitution. In unambiguous notation, we consider paired substitutions along two branches with the same specific state (spe), different states (dif) or any state (any) at the ancestral and derived nodes. The five combinatorial states that are frequently considered in the literature are paired substitutions (any→any), double divergence (any→dif), convergence (any→spe), discordant convergence (dif→spe) and congruent convergence (spe→spe) (Supplementary Fig. 1c). Convergence is discussed throughout this report because it is of particular importance in testing evolutionary genotype–phenotype associations.

## Ancestral state reconstruction and parameter estimation

Our method estimates convergent substitution via ancestral reconstruction. Whereas ancestral amino acid reconstruction has been used in previous reports[10,11,17,18], here we used codon sequence reconstruction. Using the input phylogenetic tree and observed codon sequences, CSUBST internally uses IQ-TREE to estimate the posterior probabilities (PP) of ancestral sequences by the empirical Bayesian method[80] with the ECMK07+F+R4 model by default. At the same time, the parameters used in CSUBST are estimated: equilibrium frequencies of codon state $i$ ($\pi_i$), ASRV for a codon site $l$ ($r_l$), non-synonymous per synonymous substitution ratio ($\omega$), and transition per transversion substitution ratio ($\kappa$).

## Multidimensional array structures for substitution history

CSUBST stores the coding sequences and the reconstructed probable ancestral states in a three-dimensional array whose size is $M \times L \times 61$ for a phylogenetic tree with $M$ nodes (excluding the root node) generated from an alignment of coding sequences with $L$ codon sites, each of which can take a distribution of 61 different codon states (in the universal genetic code), excluding stop codons. We denote by $P_{mlj}(X|D,\theta)$ the PP of codon $X$ for codon state $j$ at site $l$ on node $m$. The three-dimensional array for codon states is then converted to a four-dimensional array that stores the probability of substitutions with the size of $B \times L \times 61 \times 61$, where $B$ denotes the number of branches excluding the root branch. This array stores the PP of substitution $P_{blij}(S|D,\theta)$ for single substitution $S$ from ancestral codon state $i$ to derived codon state $j$ for a codon site $l$ in branch $b$. For a site $l$ in branch $b$ connecting ancestral node $n$ with codon state $i$ and descendant node $m$ with codon state $j$, the PP of substitution matrix $P_{ij}(S|D,\theta)$ is derived as

$$P_{ij}(S|D,\theta) = P_i(X|D,\theta) \times P_j(X|D,\theta)$$

$$= \begin{pmatrix} i_1 \\ i_2 \\ \vdots \\ i_{61} \end{pmatrix} \times (j_1 \ j_2 \ \cdots \ j_{61}) = \begin{pmatrix} i_1j_1 & i_1j_2 & \cdots & i_1j_{61} \\ i_2j_1 & i_2j_2 & \cdots & i_2j_{61} \\ \vdots & \vdots & \ddots & \vdots \\ i_{61}j_1 & i_{61}j_2 & \cdots & i_{61}j_{61} \end{pmatrix}. \quad (1)$$

As the transition between the same codon state is not considered a substitution, the diagonal elements ($ij_{i=j}$) are filled with 0. Although equation (1) is an approximation that does not take into account the non-independence between nodes of a phylogenetic tree, we confirmed that the effect was negligible (Supplementary Text 15 and Supplementary Fig. 14). For efficient processing of non-synonymous and synonymous substitution probabilities with the array operation of NumPy[81], the four-dimensional array is converted into a pair of five-dimensional arrays ($A^N$ and $A^S$ for non-synonymous and synonymous substitutions, respectively) whose individual size is $B \times L \times G \times I \times J$, where codon states are grouped into $G$ categories (Supplementary Fig. 2a). Stored values range between 0 and 1, denoted by $P_{blgij}(S|D,\theta)$, the probability of single substitution $S$ from ancestral codon $i$ to derived codon $j$ ($i \neq j$) in codon group $g$ at site $l$ of branch $b$, given the observed sequence data $D$ and model parameters $\theta$ that include the phylogenetic tree. The elements in the array $A^N$ dictate $P_{blgij}(S^N|D,\theta)$, the probabilities of non-synonymous substitutions ($S^N$), whereas those in the array $A^S$ correspond to $P_{blgij}(S^S|D,\theta)$, the probabilities of synonymous substitutions

($S^S$). In $A^N$, where the PPs of synonymous codons are merged, a single $20 \times 20$ matrix records all the substitution probabilities, and therefore $G = 1$ and $I = J = 20$. Synonymous substitutions occur only between codons that code for the same amino acid. Because there are 20 different amino acids, $G$ equals 20 in $A^S$. In the case of the universal genetic code, the maximum number of codons encoding the same amino acid is six, for leucine, serine and arginine, so $I = J = 6$. In the matrix corresponding to these three amino acids, all values are between 0 and 1, but for amino acids with a smaller number of codons, the out-of-range indices are filled with zero. Missing sites in the sequence alignment are also treated as zero. For simplicity, we explain the case where there is no missing site in the observed sequences and ancestral states in the following sections, but the implementation in CSUBST appropriately takes into account the missing sites by subtracting its numbers from $L$ at every necessary step in individual branches or branch combinations.

## Tree rescaling

During the ancestral state reconstruction, IQ-TREE estimates the branch length as the number of nucleotide substitutions per codon site. Because our model requires the number of codon substitutions rather than the number of nucleotide substitutions, and because branch lengths are required separately for both synonymous and non-synonymous substitutions, we obtained rescaled branch length $t_b$ of branch $b$ from the substitution probabilities as follows:

$$t_b = \frac{\sum_{l=1}^{L}\sum_{g=1}^{G}\sum_{i=1}^{I}\sum_{j=1}^{J}P_{blgij}(S|D,\theta)}{L}. \quad (2)$$

$t_b{}^N$ and $t_b{}^S$ for non-synonymous and synonymous substitutions were obtained with $P_{blgij}(S^N|D,\theta)$ and $P_{blgij}(S^S|D,\theta)$, respectively. For example, the total branch lengths of the vertebrate phosphoglycerol kinase tree before and after rescaling are 7.57 nucleotide-substitutions/codon-site and 7.21 codon-substitutions/codon-site (1.59 non-synonymous and 5.62 synonymous codon substitutions per codon site).

## Observed number of combinatorial substitutions

The only true observations are the gene sequences of the extant species, and the PPs of ancestral sequences and codon substitutions are estimates. However, we refer to the PPs as 'observations'[18] to unambiguously distinguish them from the expected values described in the next section. Here we denote by $P_l(S_C|D,\theta)$ the probability of combinatorial substitution $S_C$ at codon site $l$ given observed sequences $D$ and model $\theta$. The probabilities of non-synonymous and synonymous combinatorial substitutions at site $l$ are separately obtained as $P_l(S_C^N|D,\theta)$ and $P_l(S_C^S|D,\theta)$, respectively, with the following equations:

$$P_l^{\mathrm{any}\to\mathrm{any}}(S_C|D,\theta) = \sum_{g=1}^{G}\prod_{k=1}^{K}\sum_{i=1}^{I}\sum_{j=1}^{J}P_{klgij}(S|D,\theta) \text{ for paired substitutions,} \quad (3)$$

$$P_l^{\mathrm{any}\to\mathrm{spe}}(S_C|D,\theta) = \sum_{g=1}^{G}\sum_{\substack{j=1 \\ j_{k_1}=j_{k_2}}}^{j}\prod_{k=1}^{K}\sum_{i=1}^{I}P_{klgij}(S|D,\theta) \text{ for convergence,} \quad (4)$$

and

$$P_l^{\mathrm{spe}\to\mathrm{spe}}(S_C|D,\theta) =$$
$$\sum_{g=1}^{G}\sum_{\substack{i=1 \\ i_{k_1}=i_{k_2}}}^{I}\sum_{\substack{j=1 \\ j_{k_1}=j_{k_2}}}^{J}\prod_{k=1}^{K}P_{klgij}(S|D,\theta) \text{ for concordant convergence,} \quad (5)$$

where $k$ represents a branch of interest. We denote by $K$ the degree of combinatorial substitutions or the number of branches to be compared. Because two branches are often compared in conventional convergence analysis, we explain here the case of $K = 2$. A part of array

operations in equations (3)–(5) are illustrated in Supplementary Fig. 2b. The total probabilities of observed substitution pairs across sites in the branch pair are calculated as

$$O_C = \sum_{l=1}^{L} P_l(S_C|D,\theta). \tag{6}$$

$O_C$ is separately obtained for non-synonymous and synonymous combinatorial substitutions ($O_C^N$ and $O_C^S$, respectively). By definition (Supplementary Fig. 1c), the values of $O_C$ for double divergence and discordant convergence are derived as follows at $K = 2$:

$$O_C^{\mathrm{any}\to\mathrm{dif}} = O_C^{\mathrm{any}\to\mathrm{any}} - O_C^{\mathrm{any}\to\mathrm{spe}} \text{ for double divergence} \tag{7}$$

and

$$O_C^{\mathrm{dif}\to\mathrm{spe}} = O_C^{\mathrm{any}\to\mathrm{spe}} - O_C^{\mathrm{spe}\to\mathrm{spe}} \text{ for discordant convergence.} \tag{8}$$

$C/D$[17] corresponds to $O_N^{\mathrm{any}\to\mathrm{spe}}/O_N^{\mathrm{any}\to\mathrm{dif}}$ in our notation.

### Expected number of combinatorial substitutions

To estimate the rate of combinatorial substitutions, the observed number $O_C$ is contrasted with the expected number $E_C$. $E_C$ is derived from codon substitution models in a way similar to the previous application of amino acid substitution models[18]. The tested codon substitution models include the empirical models ECMK07 and ECMrest[74] and the mechanistic models MG[82] and GY[83]. The same model was consistently used in the ancestral state reconstruction and in deriving the model-based expectations of combinatorial substitutions. In the method described below, empirical equilibrium codon frequencies, the rescaled branch length and ASRV are also taken into account. In the empirical models, the codon substitution rate matrix $Q$ is derived according to previous literature[74,84] as follows:

$$Q = \{q_{ij}\} = \begin{pmatrix} - & s_{1,2} & \cdots & s_{1,61} \\ s_{2,1} & - & \cdots & s_{2,61} \\ \vdots & \vdots & \ddots & \vdots \\ s_{61,1} & s_{61,2} & \cdots & - \end{pmatrix} \times \mathrm{diag}\left(\pi_1 \ \pi_2 \ \cdots \ \pi_{61}\right), \tag{9}$$

where $s_{i,j}$ denotes the exchangeabilities of codon pairs $i$ and $j$ ($s_{ij} = s_{ji}$) and $\pi_i$ represents the equilibrium frequencies of 61 codons estimated from the input alignment. In the mechanistic models, mechanistic substitution parameters are used instead of the exchangeabilities. In the MG model, $q_{ij}$ is obtained with $\pi_i$ and non-synonymous per synonymous substitution ratio $\omega$, whereas transition per transversion substitution ratio $\kappa$ is also taken into account in the GY model. $Q$ is then rescaled as

$$\sum_{i=1}^{61}\sum_{\substack{j=1 \\ j\neq i}}^{61} \pi_i q_{ij} = 1. \tag{10}$$

Finally, the diagonal elements of $Q$ are completed as

$$q_{ii} = -\sum_{\substack{j=1 \\ j\neq i}}^{61} q_{ij}. \tag{11}$$

With substitution rate $r_l$ pre-estimated by IQ-TREE, the codon transition probability matrix $P_{ij}(t_b,r_l)$ after time $t_b$ are obtained using matrix exponentiation as

$$P_{ij}(t_b,r_l) = e^{Qt_b r_l}, \tag{12}$$

where CSUBST uses the rescaled branch lengths $t_b^N$ or $t_b^S$ in place of $t_b$. The distribution of expected substitutions at site $l$ in branch $b$ connecting ancestral node $n$ with codon state $i$ and a descendant node is therefore given by

$$P_{ij}(S^{\mathrm{expected}}|D,\theta) = P_i(X|D,\theta) \times P_{ij}(t_b,r_l). \tag{13}$$

Using $P_{klgij}(S^{\mathrm{expected}}|D,\theta)$ in place of $P_{klgij}(S|D,\theta)$, the total probabilities of expected substitution pairs across sites in the branch pair denoted by $E_C$ are obtained by the same procedure used to obtain $O_C$ (equations (3)–(8)). Similar to $O_C$, the expected numbers of combinatorial substitutions ($E_C$) are separately calculated for non-synonymous and synonymous substitutions ($E_C^N$ and $E_C^S$, respectively). By definition (Supplementary Fig. 1c), the following relationships hold at $K = 2$:

$$E_C^{\mathrm{any}\to\mathrm{dif}} = E_C^{\mathrm{any}\to\mathrm{any}} - E_C^{\mathrm{any}\to\mathrm{spe}} \tag{14}$$

and

$$E_C^{\mathrm{dif}\to\mathrm{spe}} = E_C^{\mathrm{any}\to\mathrm{spe}} - E_C^{\mathrm{spe}\to\mathrm{spe}}. \tag{15}$$

### Non-synonymous and synonymous combinatorial substitution rates

With the observed and expected numbers of combinatorial substitutions ($O_C$ and $E_C$, respectively), the rates of non-synonymous and synonymous combinatorial substitutions are obtained, respectively, by

$$dN_C = O_C^N/E_C^N \tag{16}$$

and

$$dS_C = O_C^S/E_C^S. \tag{17}$$

Both $dN_C$ and $dS_C$ are the observed number divided by the expected number. If the theoretically expected number derived from the codon substitution model fully explains the observed number, then both $dN_C$ and $dS_C$ values would be 1.0. $dN_C$ can be regarded as equivalent to $R$ with the per-gene equilibrium amino acid frequencies (their $f_{\mathrm{gene}}$), but note that some features are different from the corresponding parts for $R$. In particular, we used the standard procedure to derive codon transition probabilities (equations (12) and (13) and equation 1.2 in ref. [16]), whereas no matrix exponentiation is applied for $R$. In the 21 vertebrate genome dataset, the total expected convergence ($E_C^{N,\mathrm{any}\to\mathrm{spe}} = 6{,}051{,}985$) corresponds to 87.2% of the total observed convergence ($O_C^{N,\mathrm{any}\to\mathrm{spe}} = 6{,}939{,}070$). This expectation matches the observation with better accuracy than the previously published results with the *Drosophila* genomes ($582.8/942 = 61.9\%$ with their JTT-$f_{\mathrm{gene}}$ model)[18].

### Accounting for a range of combinatorial substitution rates

Under purifying selection, which is the default evolutionary mode of many proteins, the rate of synonymous substitutions is faster than that of non-synonymous substitutions. Therefore, saturation of synonymous substitutions becomes a potential problem, especially in a counting method that cannot properly account for the effects of multiple substitutions. To account for this issue, we applied a transformation of $dS_C$ using quantile values ($U_p$) as follows:

$$dS_C^{\mathrm{corrected}} = \begin{cases} dS_C^{\mathrm{uncorrected}}, \text{if } dS_C^{\mathrm{uncorrected}} \geq dN_C \\ U_{p^{dS_C}}^{dN_C}, \text{otherwise} \end{cases}, \tag{18}$$

where $U_{p^{dS_C}}^{dN_C}$ denotes the quantile value of the empirical $dN_C$ distribution at $p^{dS_C}$, the quantile rank of the $dS_C$ value, among all branch combinations at $K$. This operation rescales $dS_C$ to match its distribution range

with that of $dN_C$, and the resulting $\omega_C$ becomes robust for outlier values (Supplementary Fig. 15). Because of the need for quantile values, this transformation is only applicable when the branch combinations are exhaustively searched. In this work, $dS_C^{corrected}$ is used at $K = 2$ unless otherwise mentioned.

### Non-synonymous per synonymous combinatorial substitution rate ratio

A non-synonymous per synonymous combinatorial substitution rate ratio for $K$ branches is given by

$$\omega_C = \frac{dN_C}{dS_C} = \frac{O_C^N/E_C^N}{O_C^S/E_C^S}. \tag{19}$$

$\omega_C$ can be separately calculated for different categories of combinatorial substitutions, for example, $\omega_C^{any \to any}$ for paired substitutions, $\omega_C^{any \to spe}$ for double divergence, $\omega_C^{any \to dif}$ for convergence, $\omega_C^{dif \to spe}$ for discordant convergence and $\omega_C^{spe \to spe}$ for concordant convergence. For simplicity, the derivation of $\omega_C$ was explained above for the combinatorial substitutions illustrated in Supplementary Fig. 1c. However, our method can be applied to other categories of combinatorial substitutions as well. For example, phenotypic convergence may be associated with the same ancestral amino acid substituted to different amino acids[85], in which case $\omega_C^{spe \to any}$ may be useful for analysis.

### Branch combinations

Combinatorial substitutions are a collection of independently occurring evolutionary events (Supplementary Fig. 1c). Branch combinations containing an ancestor–descendant relationship did not satisfy the evolutionary independence and were therefore excluded from the analysis. Although convergent substitutions occurring in sister branch pairs satisfy the evolutionary independence, they are difficult to discriminate and are often treated as a single ancestral substitution. For this reason, sister branches were also excluded from the analysis (Supplementary Fig. 12a).

### Analysis of higher-order branch combinations

$O_C$ and $E_C$, and hence $\omega_C$, can also be obtained for combinations of more than two branches ($K > 2$). The higher-order analysis is particularly useful when analysing traits with extensively repetitive convergence, such as $C_4$ photosynthesis, which is thought to have evolved at least 62 times independently[86]. To efficiently explore the higher-order dimensions of branch combinations, we devised a branch-and-bound algorithm that combines the convergence metric cut-off, and the generation of $K + 1$ branch combinations from the branch overlaps at $K − 1$ (Fig. 4a and Supplementary Fig. 12a). The higher-order analysis starts with an exhaustive comparison of branch pairs (that is, $K = 2$). Next, convergent branch pairs are extracted with an $\omega_C$ cut-off value ($\geq 5.0$ in Fig. 4). At this time, branch pairs with a small number of convergent substitutions are excluded by applying an $O_C^N$ cut-off value ($\geq 2.0$ in Fig. 4). The convergent branch pairs are then subjected to the all-versus-all comparison. When a shared branch is found, their union is generated as a combination of three branches to be analysed. Before proceeding to the analysis at $K = 3$, branch combinations containing a sister or ancestor–descendant relationship are discarded. In this way, $K$ is sequentially increased by one at a time. As such, the algorithm searches only for higher-order branch combinations that are guaranteed to have sufficient convergence metrics in lower-order combinations. In each round, convergent branch combinations are first extracted by the cut-offs, and then the $K + 1$ combinations are generated by the $K − 1$ overlap, as in the analysis at $K = 2$. For example, two, three and four branches should be shared at $K = 3$, $K = 4$ and $K = 5$, respectively. The increase in $K$ continues until the algorithm no longer finds a branch combination that satisfies the criteria of $\omega_C$ and $O_C^N$.

### Implementation of CSUBST

The proposed methods, including the calculation of $\omega_C$ and the branch-and-bound algorithm for higher-order combinations, were implemented in the 'analyze' function of CSUBST, which was written in Python 3 (https://www.python.org/). Phylogenetic tree processing was implemented with the Python package ETE 3 (ref. [87]). Numpy[81], SciPy[88] and pandas (https://pandas.pydata.org/) were used for array and table data processing. Parallel computation was performed by multiprocessing with Joblib (https://joblib.readthedocs.io/en/latest/). The intensive calculation was optimized with Cython[89].

### Mapping combinatorial substitutions to protein structures

For the analysis of protein structures, a streamlined pipeline was implemented in the 'site' function of CSUBST. Using the '--pdb besthit' option, CSUBST requests an online MMseqs2 search[90] against the RSCB Protein Data Bank (PDB)[91] to obtain three-dimensional conformation data of closely related proteins. If no hit is obtained, a BLASTP search against the UniProt database is run on the QBLAST server to identify the best hit protein for which AlphaFold-predicted structure is available[92,93]. For some proteins, structural data were manually selected because more appropriate structures were available (for example, with substrate). Subsequently, CSUBST internally uses MAFFT to generate protein alignments to determine the homologous positions of amino acids and to write a PyMOL session file. The protein structures were visualized using Open-Source PyMOL v2.4.0 (https://github.com/schrodinger/pymol-open-source).

### Reporting summary

Further information on research design is available in the Nature Portfolio Reporting Summary linked to this article.

## Data availability

Raw data and results are available at https://doi.org/10.5061/dryad.tx95x6b0v[94].

## Code availability

CSUBST is available from GitHub (https://github.com/kfuku52/csubst). The results reported in this study can be reproduced with CSUBST v0.20.17. The notation in this paper is consistent with CSUBST v1.0.0. Scripts used in this study are available at https://doi.org/10.5061/dryad.tx95x6b0v[94].

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

## Acknowledgements

We acknowledge the following sources for funding: MEXT/JSPS KAKENHI 18J00178 (K.F.), the Sofja Kovalevskaja programme of the Alexander von Humboldt Foundation (K.F.), a Human Frontier Science Program Young Investigators Grant RGY0082/2021 (K.F.) and National Institutes of Health (NIH) R01 GM083127 (D.D.P.). Computations were partially performed on the National Institute of Genetics supercomputer.

## Author contributions

K.F. conceptualized the study. K.F. designed and wrote all programmes and performed data analysis. D.D.P. contributed to conceptualizing and helping guide the analysis. K.F. and D.D.P. wrote the paper.

## Competing interests

The authors declare no competing interests.

## Additional information

**Correspondence and requests for materials** should be addressed to Kenji Fukushima.

# Reporting Summary

## Statistics

For all statistical analyses, confirm that the following items are present in the figure legend, table legend, main text, or Methods section.

| n/a | Confirmed | |
|---|---|---|
| ☐ | ☒ | The exact sample size (*n*) for each experimental group/condition, given as a discrete number and unit of measurement |
| ☐ | ☒ | A statement on whether measurements were taken from distinct samples or whether the same sample was measured repeatedly |
| ☐ | ☒ | The statistical test(s) used AND whether they are one- or two-sided *Only common tests should be described solely by name; describe more complex techniques in the Methods section.* |
| ☒ | ☐ | A description of all covariates tested |
| ☐ | ☒ | A description of any assumptions or corrections, such as tests of normality and adjustment for multiple comparisons |
| ☐ | ☒ | A full description of the statistical parameters including central tendency (e.g. means) or other basic estimates (e.g. regression coefficient) AND variation (e.g. standard deviation) or associated estimates of uncertainty (e.g. confidence intervals) |
| ☐ | ☒ | For null hypothesis testing, the test statistic (e.g. *F*, *t*, *r*) with confidence intervals, effect sizes, degrees of freedom and *P* value noted *Give P values as exact values whenever suitable.* |
| ☒ | ☐ | For Bayesian analysis, information on the choice of priors and Markov chain Monte Carlo settings |
| ☒ | ☐ | For hierarchical and complex designs, identification of the appropriate level for tests and full reporting of outcomes |
| ☐ | ☒ | Estimates of effect sizes (e.g. Cohen's *d*, Pearson's *r*), indicating how they were calculated |

*Our web collection on statistics for biologists contains articles on many of the points above.*

## Software and code

Policy information about availability of computer code

| Data collection | No software was used. |
|---|---|
| Data analysis | BUSCO v4.0.5, OrthoFinder v2.4.1, Trinity v2.8.5, fastp v0.20.1, TransDecoder v5.5.0, BLAST+ v2.9.0, TMHMM v2.0, Trinotate v3.2.1, MAFFT v7.455, EMBOSS v6.6.0, ClipKIT v0.1.2, IQ-TREE v2.0.3, PAML v4.9, GeneRax v1.2.2, l1ou v1.40, Open-Source PyMOL v2.4.0, RADTE v6957377, CDSKIT v0.9.1, NWKIT v0.10.0, CSUBST v.0.20.17, and custom scripts deposited to Dryad (https://doi.org/10.5061/dryad.tx95x6b0v). |

For manuscripts utilizing custom algorithms or software that are central to the research but not yet described in published literature, software must be made available to editors and reviewers. We strongly encourage code deposition in a community repository (e.g. GitHub). See the Nature Portfolio guidelines for submitting code & software for further information.

## Data

Policy information about availability of data

All manuscripts must include a data availability statement. This statement should provide the following information, where applicable:
- Accession codes, unique identifiers, or web links for publicly available datasets
- A description of any restrictions on data availability
- For clinical datasets or third party data, please ensure that the statement adheres to our policy

Raw data and results are available in the Supplementary Dataset (https://doi.org/10.5061/dryad.tx95x6b0v).

# Field-specific reporting

Please select the one below that is the best fit for your research. If you are not sure, read the appropriate sections before making your selection.

☒ Life sciences          ☐ Behavioural & social sciences          ☐ Ecological, evolutionary & environmental sciences

For a reference copy of the document with all sections, see nature.com/documents/nr-reporting-summary-flat.pdf

# Life sciences study design

All studies must disclose on these points even when the disclosure is negative.

| | |
|---|---|
| Sample size | We sampled as many taxa as or more than previously individually reported for the analysis of known protein convergence. In the genome-wide analysis, we used a set of 21 animal genomes for which we previously confirmed that expression evolution was analyzable with the OU modeling. |
| Data exclusions | No data were excluded from the analysis. |
| Replication | All results presented in this paper can be reproduced from the data provided as Supplementary Data. |
| Randomization | Randomization is not relevant because we did not conduct experiments where randomization was applicable. |
| Blinding | Blinding is not relevant to our study. |

# Reporting for specific materials, systems and methods

We require information from authors about some types of materials, experimental systems and methods used in many studies. Here, indicate whether each material, system or method listed is relevant to your study. If you are not sure if a list item applies to your research, read the appropriate section before selecting a response.

## Materials & experimental systems

| n/a | Involved in the study |
|---|---|
| ☒ | ☐ Antibodies |
| ☒ | ☐ Eukaryotic cell lines |
| ☒ | ☐ Palaeontology and archaeology |
| ☒ | ☐ Animals and other organisms |
| ☒ | ☐ Human research participants |
| ☒ | ☐ Clinical data |
| ☒ | ☐ Dual use research of concern |

## Methods

| n/a | Involved in the study |
|---|---|
| ☒ | ☐ ChIP-seq |
| ☒ | ☐ Flow cytometry |
| ☒ | ☐ MRI-based neuroimaging |

