## [Peer Review File. · Nature Ecology & Evolution]

Peer Review Information

Journal: Nature Ecology & Evolution

Manuscript Title: Detecting macroevolutionary genotype-phenotype associations using error-corrected rates of protein convergence

Corresponding author name(s): Kenji Fukushima

Editorial Notes:

Reviewer Comments & Decisions:

Decision Letter, initial version:
--

6th May 2022

Dear Dr Fukushima,

Your manuscript entitled "Detecting macroevolutionary genotype-phenotype associations using error-corrected rates of protein convergence" has now been seen by 2 reviewers, whose comments are attached. The reviewers have raised a number of concerns which will need to be addressed before we can offer publication in Nature Ecology & Evolution. We will therefore need to see your responses to the criticisms raised and to some editorial concerns, along with a revised manuscript, before we can reach a final decision regarding publication.

We therefore invite you to revise your manuscript taking into account all reviewer and editor comments. Please highlight all changes in the manuscript text file.

* Include a "Response to reviewers" document detailing, point-by-point, how you addressed each

reviewer comment. If no action was taken to address a point, you must provide a compelling argument. This response will be sent back to the reviewers along with the revised manuscript.

* If you have not done so already please begin to revise your manuscript so that it conforms to our Article format instructions at <http://www.nature.com/natecolevol/info/final-submission>. Refer also to any guidelines provided in this letter.

[REDACTED]

Nature Ecology & Evolution is committed to improving transparency in authorship. As part of our efforts in this direction, we are now requesting that all authors identified as 'corresponding author' on published papers create and link their Open Researcher and Contributor Identifier (ORCID) with their account on the Manuscript Tracking System (MTS), prior to acceptance. ORCID helps the scientific community achieve unambiguous attribution of all scholarly contributions. You can create and link your ORCID from the home page of the MTS by clicking on 'Modify my Springer Nature account'. For more information please visit <http://www.springernature.com/orcid>.

[REDACTED]

Reviewers' comments:

Reviewer #1 (Remarks to the Author):

2This manuscript presents a new method for detecting the presence of adaptive protein convergence. Given the issues with many previous methods, I thought this was an ingenious solution (and a rigorous one). I thought the many included analyses made a convincing argument that the method was fairly robust to false positives, and that it avoids many of the problems faced by previous measures of convergence. I predict that many people will use it.

That being said, I do have some comments that I hope can help to improve the manuscript. A couple of these are technical, but a couple are conceptual, too. I list a few more minor comments at the end.

1. Statistical power. While there is a lot of focus on reducing false positives, I could find very little on reducing false negatives, or the statistical properties of ω_C overall. So even though the Discussion says that ω_C doesn't have any reduction in power (line 489), I didn't know what this was referring to. There is one small mention of "true positives" in Figure 1D (and Figure S3A), but it's not clear what this shows. Even a small simulation study of the number of convergent substitutions necessary to detect significance would be helpful.

But this just begs the question: how does a researcher know when ω_C is significant? I saw nothing that provided guidelines for this. In some places the authors use the top 1% of ω_C values to identify genes of interest, while in Figure 1D they seem to use neutral simulations to call "true positives". Some clarification of the recommended usage of the statistic would be extremely useful.

2. Genotype-phenotype associations. I have to admit that I could have done without claims about genotype-phenotype associations altogether. In some contexts, the claims are pretty empty (e.g. line 67), but in most contexts the results do not support the conclusions at all. For instance, the "identification of molecular convergence associated with a particular phenotype" (between ruminants and rabbits) identified 352 "candidate branch pairs". So which are the genes involved in this phenotype? (And which phenotype, exactly? Herbivory?) While I am convinced the method found truly convergent proteins, I am not convinced that they were linked to any particular phenotype, or could be shown to be. The brief stories about a bunch of proteins were not super-convincing.

Even more worrying, the next section of the manuscript then describes how to look for convergence without a pre-specified phenotype, which will "provide a basis for understanding overlooked phenotypes". But which ones? How would we identify these phenotypes? Especially on internal branches of species trees, I don't have the foggiest idea how these signals would lead to an increased understanding of genotype-phenotype associations.

Putting aside the question of whether the method can help to identify such associations, if the authors really think this is a good use of their method, I think there are two key references missing: Smith et al. (2020, TREE) on the "PhyloG2P" approach, and Pease et al. (2016, PLoS Biology) on the "phyloGWAS" approach.

3. Convergence in duplicated genes. While I do see how a search for convergent evolution between gene duplicates would be very useful, I had a bit of trouble figuring out how this was actually done

3(there was very little description). For instance, are paralogs only in different species compared, or all pairs of paralogous branches? It seems like both might be informative, but about different biological processes. Similarly, why require two duplication events to separate paralogs ("DD")? It seems biologically more plausible that duplicates separated by only one duplication event (plus one speciation) would be more likely to converge, for the very same reasons that epistasis is invoked to explain decreasing convergence through time.

Also, how do you assign convergent substitutions to gene trees with missing genes? While this problem must also occur for single-copy genes, I imagine it must be much more common for trees with duplicates in them. And for trees with losses, it is not clear to me how branches can be compared fairly. Clearer explanations for the use of paralogs—and the patterns found (especially a bit more on the patterns described on lines 277-281)—would be helpful.

4. "Neutral" evolution. I admit this is a bit of a bugbear for me, but the use (and mis-use) of this term throughout the paper was both misleading and incorrect. Neutral evolution does not mean "no selection" or "no constraint"—it means the absence of positive selection (and the absence of balancing selection and weakly deleterious substitutions). While I know that there is a portion of the phylogenetic literature that says $\omega=1$ is evidence of "neutral evolution," this is not correct. There is nothing about the neutral theory or neutral evolution that rejects a role for constraint, and this mis-use should really not be condoned.

Even more importantly for the clarity of the arguments being made here: the authors only require the correct usage of neutral. In other words, the neutral expectation of $\omega_{C} = 1$, because it is a ratio of convergent substitutions of each type. But this expectation does not require that $\omega=1$, even though this is what is said (line 159). In fact, it seems that the neutral simulations were actually done with $\omega=0.5$ (if I'm reading Table S2 correctly), which is perfectly sensible. (It seems like ω_{C} is mostly independent of the values of ω .) The conflation of these two uses of "neutral expectations" (one of which is not correct) simply made the manuscript harder to understand.

Similarly, it is not clear to me what the line for "Theoretically Neutral" is pointing to in the panels of Figure 1D with dN_{C} and dS_{C} in them. Why should these two metrics have an expectation of 1? The citation to Bustamante et al. 2005 (line 780) was also confusing, as the McDonald-Kreitman test isn't a test for purifying selection. The results in that paper simply say that there are segregating weakly deleterious polymorphisms in humans. Finally, "nearly neutral" and "essentially nonfunctional" (line 75) are not contrasting ideas—the former is about the fitness effect of substituting one allele for another, and the latter describes a change in molecular function.

5. Minor comments.

- dN and dS are not rates, they are distances.

-I am not sure what the section on "Extracting a high-confidence set of convergent lineages" (starting line 287) shows, exactly. That there are outliers on any pair of branches?

-Could the authors comment on the relationship of the metrics they use to the one proposed in

4Burskaia et al. (2020, GBE)?

-Figure S3C seems to show quite a bit of sensitivity of ω_c to misspecification of the codon model. Maybe this should be mentioned in the main text?

- It was not clear to me that the expression changes detected are adaptive. What if there is a lot of overlap with testis expression because this pattern accumulates at high (neutral) rates?

-Figure 3 was not very informative. Maybe some more detail on the expression method/patterns, and less on protein structures?

-Is there a formula for number of independent branches in different sized trees? I'm just wondering whether there's an easy way to get the number laid out starting on line 446, as it seems like a very helpful calculation.

-The "branch-and-bound" algorithm described seems to be in fact be a greedy algorithm. I think the same approach is often taken in trying to detect pairwise epistasis in mapping studies.

[REDACTED]

Reviewer #2 (Remarks to the Author):

In this manuscript, Fukushima and Pollock develop a new method to detect protein-coding genes with an excess of convergent amino acid substitutions, that corrects for the biases and false positives that have plagued this field for a while. Their approach, implemented in the software package CSUBST, appears to be robust and performs well in the tests reported in this manuscript, making it highly likely this will be adopted as a standard method in the community of evolutionary biologists who study convergent evolution. I would further note that the code associated with the manuscript is notably well organized, easy to install, and simple to work with, with multiple example datasets and a well organized wiki/help section. Nonetheless, the manuscript would be significantly improved by some revisions, which I outline here. Of particular importance are the technical issues in point 4 below.

1. As the authors mention in their discussion (503-511), approaches such as CSUBST rely on identifying genes with more convergent substitutions than expected under some kind of neutral model; this approach can have low power when there are only one or a few convergent substitutions in a protein. While CSUBST is innovative in using convergent synonymous substitutions to normalize the excess over expectations in nonsynonymous substitutions, there is an alternate line of research, perhaps best exemplified by Marcovitz et al 2019 (PMC6800341), that instead considers the full collection of convergence substitutions and asks questions about functional enrichment of the genes affected. I would be very interested to see if there is a way that these ideas could be merged with the authors' method. For example, it seems like it could be possible to aggregate by, e.g., some kind of

5functional class instead of gene, to gain significantly more power to detect cases of adaptive convergence. This could be conceptually similar to methods like INSIGHT (Gronau et al 2013, <https://doi.org/10.1093/molbev/mst019>), which aggregate "genomically coherent elements" for tests of selection to increase power when the substitutions of interest are rare. I realize that while fully implementing this kind of extension could be beyond the scope of the current manuscript, a more extensive discussion of limitations and possible extensions of the CSUBST method would help contextualize the current work and its potential and future impact.

2. Related to this point, a challenge for many studies seeking to identify convergent amino acid substitutions is that there is the possibility that this is simply not a particularly common phenomenon. While a few key examples are repeatedly discussed (PEPC in C4 photosynthesis, prestin in echolocating mammals, ATPalpha1 in glycoside resistant insects, etc), an exciting possibility of the new method the authors propose is the potential to do exploratory analysis in the absence of known candidates, as the authors discuss in the section starting on line 344 and again the discussion, as well as showing in Fig 3 and Fig 5. However, it is not particularly clear from the results the authors' present how well this works. While I understand the motivation for focusing on the small number of genes that show overlapping patterns of convergence between gene expression and amino acid datasets, the results of this analysis seem more relevant to questions of the role of gene duplication in protein evolution than the kind of genotype-phenotype link the authors emphasize in e.g. Fig 5. Therefore, I think that a few additional analyses in this section would be particularly valuable.

i. First, the authors identify 53,805 candidate branch pairs in their exploratory analysis. This would average to about 100 cases of excess convergence for a pair of species, as the authors discuss on lines 351-356. It is notable, to me, that the number of branch pairs with excess convergence in the herbivore analysis is substantially higher than this average, at 352. This raises the question of what the distribution of these number actually are. Do species pairs with well-known convergent phenotypes tend to have higher-than-average numbers of genes with excess convergence? The general conclusion in many previous studies has been that this is not the case, but given that previous methods were very sensitive to false positives this could swamp real signal.

ii. Second, from the analysis of PEPC in C4 photosynthesis plants, it seems like a genome-wide, unbiased screen for say 3- or 4-way convergence in the mammalian tree (something that has not really been possible before CSUBST) could be extremely informative to identify proteins that are highly likely to be biological relevance for interesting phenotypes. This kind of analysis would go a long way towards demonstrating the clear potential of CSUBST for the kinds of exploratory analyses the authors pitch as one of the impactful areas of study enabled by their work.

3. A minor point about the CSUBST program. While the documentation for installation and basic usage is admirably clear, and the examples for advanced usage are nice to see, the program has a huge number of options that are not clear how to set or use, many of which I presume the user will rarely, if ever, want to modify. A short addition to the readme that describes the submodules (dataset, analyze, simulate, site) with a brief description of the options the user is most likely to want to change, would be extremely helpful.

4. I also have some technical questions that should be addressed.

6i. The authors use a variety of $\omega[c]$ and $\text{observed}[N][c]$ thresholds to identify genes with excess convergence, and these thresholds vary between the higher-order branch-and-bound search and the exhaustive pairwise search. However, there is little/no discussion in the paper of why these thresholds were chosen (unless I missed it, in which case I apologize). I think this really needs to be clearly explained and justified. Is there some reason to believe that an $\omega[c]$ value of 3 is a particularly important cutoff? Is this what seems to work in practice, or is there a statistical justification for it? Are the same cutoffs likely to be universally useful regardless of tree size and species included, or is this something the user is likely going to have to optimize? Of particular concern is the fact that several of the key examples presented in Fig 1 / Table S4 (e.g., *prestin* with $O_C^N \sim 1.09$, *ATPalpha1* with $O_C^N \sim 1.4$) don't seem to pass these same thresholds. The simulation data shows the advantages of $\omega[c]$ clearly, but it is less obvious, in practice, how one translates this to real data. The method seems fast enough that adding some kind of simulation or permutation based P-value may be of significant value and may need to be explored to address some of these issues.

ii. After digging into the results in Tables S5 and S6, I am a little bit confused about how large gene families with speciation and duplication events are being analyzed. For example, looking at OG0000039, there are four branch pairs with excess convergence. Both 155/202 and 155/209 are listed as 'SS', as in both branches are speciation events. However, it seems to me that 202 and 209 must represent different paralogs, right? So, I think you have convergence between paralog A[rabbit] and paralog A[ruminant], as well as convergence between paralog A[rabbit] and paralog B[ruminant]. Something similar seems to happen in OG0000059. I don't really understand in this case why all of these are being listed as 'SS' branching types. I think a supplemental figure or diagram would be very helpful here to clarify what is going on. Additionally, I think it would be extremely useful to have a way to filter to identify convergent events only between branches where the most recent common ancestor of the two convergent branches is a speciation event.

iii. Related to the last point, I am in general somewhat confused as to how gene duplication is treated in this manuscript. Looking at Figure 1C, it seems for the simplified tree ((species 1 copy 1, species 2 copy 1),(species 1 copy 2, species 2 copy 2)) convergence between copy 1 in species 1 and copy 2 in species 1 would be treated as a 'SS' event, since the immediately preceding node in each case is a speciation event. I am not sure if this really makes sense? It would perhaps be more intuitive to classify branch pairs by whether their common ancestor is a speciation or a duplication event, especially as in the current framework it is not obvious at all what an 'SD' event means.

*****END*****

Author Rebuttal to Initial comments

Response to reviewer comments

We thank the reviewers for their thorough and thoughtful reviews. All criticisms are addressed in a point-by-point fashion below. We believe this has resulted in an improved manuscript that is appropriate for a *Nature Ecology & Evolution* audience.

Reviewer #1

Comment 1.0.0. This manuscript presents a new method for detecting the presence of adaptive protein convergence. Given the issues with many previous methods, I thought this was an ingenious solution (and a rigorous one). I thought the many included analyses made a convincing argument that the method was fairly robust to false positives, and that it avoids many of the problems faced by previous measures of convergence. I predict that many people will use it. That being said, I do have some comments that I hope can help to improve the manuscript. A couple of these are technical, but a couple are conceptual, too. I list a few more minor comments at the end.

Response: We thank the reviewer for their helpful comments and appreciation of the technical advance of this work.

Comment 1.1.0. Statistical power. While there is a lot of focus on reducing false positives, I could find very little on reducing false negatives, or the statistical properties of ω_C overall. So even though the Discussion says that ω_C doesn't have any reduction in power (line 489), I didn't know what this was referring to. There is one small mention of "true positives" in Figure 1D (and Figure S3A), but it's not clear what this shows. Even a small simulation study of the number of convergent substitutions necessary to detect significance would be helpful.

Response: In response to this comment, we added a new Supplementary Figure that shows the relationships between the observed number of nonsynonymous convergence and true positive rates.

Change in main text: Using the distribution of metric values under the Neutral scenario as a reference, we see that 70–80% of the detection metric values in the Convergent scenario are above the 95th percentile of the 1,000 simulations in their respective neutral distributions, while only 3.5% of dS_C values are above this threshold, indicating that this level of convergence is usually detected by all three of the protein convergence metrics (Fig. 1D). In ω_C , this level of detection was achievable with only two to three nonsynonymous convergent substitutions, and the positive rates exceeded 95% with seven or more convergent substitutions (Fig. S3).

Change in Supplementary Figure:

Figure S3. The true positive rate increases with the number of convergent substitutions. Simulated data analyzed were from the Convergent scenario of Fig. 1D.

Comment 1.1.1. But this just begs the question: how does a researcher know when ω_C is significant? I saw nothing that provided guidelines for this. In some places the authors use the top 1% of ω_C values to identify genes of interest, while in Figure 1D they seem to use neutral simulations to call “true positives”. Some clarification of the recommended usage of the statistic would be extremely useful.

Response: In response to this and other comments, we added a new paragraph in the Discussion as follows.

Change: It is useful to provide some clarification of our recommended usage of the ω_C statistic, particularly because in this manuscript we used different thresholds, depending on the context, to characterize ω_C values that indicate genes of interest. In the simulation analyses, we used the neutral simulations (1,000 replications) to define false positives with the 95th percentile threshold in test simulations (Fig. 1D). Because these are simulations, we know the number of true positives, and can also calculate the true positive rate distribution across simulations (Fig. S3). While this provides a useful theoretical guide to the behavior of the statistic, it should be recognized that sites were independent of each other in these simulations. In real proteins, epistasis may affect the generation of both true and false positives. Thermodynamic simulations of protein evolution (Goldstein and Pollock, 2017) may be able to overcome this problem but would require a large number of costly computational simulations, and still, the improvement in predictive value for real data would not be certain and would require extensive validation that is beyond the scope of this paper. Permutations are an alternative approach to obtaining false-positive convergence estimates, but we caution that it is not certain how substitutions should be randomized in the context of epistasis and among sites with varying constraints and substitution rates; again, any such approach would make uncertain improvements and would require extensive validation. To manage this uncertainty in our real data analysis, we compared the top 1% of values among methods (a rank-based threshold) to allow a fair comparison of different convergence metrics within the same dataset (i.e., C/D , dN_C , or ω_C). In genome-scale analyses, such rank-based thresholds can extract the most promising convergent branch combinations among a large number of observations. For particular gene families or specific lineage combinations, the number of observations is often small and it is useful to choose a reasonable threshold based on our analyses so far. For the animal genome analysis, we used a threshold of ω_C greater than 3.0 that corresponds to the 92nd percentile of all analyzed branch pairs with more than three nonsynonymous convergence ($O_C^N \geq 3.0$), while in the higher-order analysis of PEPC we employed a more stringent threshold ($\omega_C > 5.0$) to control for the combinatorial explosion. The threshold setting for ω_C should be considered and potentially adjusted based on the above caveats and the needs of a given research project, but ultimately the utility of convergence analysis will depend on the validated utility of these predictions across a variety

of biological contexts, and we recommend these thresholds as reasonable standardized starting points based on our analyses.

Comment 1.2.0. Genotype-phenotype associations. I have to admit that I could have done without claims about genotype-phenotype associations altogether. In some contexts, the claims are pretty empty (e.g. line 67), but in most contexts the results do not support the conclusions at all. For instance, the “identification of molecular convergence associated with a particular phenotype” (between ruminants and rabbits) identified 352 “candidate branch pairs”. So which are the genes involved in this phenotype? (And which phenotype, exactly? Herbivory?) While I am convinced the method found truly convergent proteins, I am not convinced that they were linked to any particular phenotype, or could be shown to be. The brief stories about a bunch of proteins were not super-convincing.

Response: We thank the reviewers for this comment. We realized that we were not clear enough about the context of this claim. Briefly, it is thought that specific genotypic changes are linked to specific phenotypic changes through their effect on the function of the genes in which the genotypic change occurs at the time the change occurs. It is unlikely that such genotype-phenotype associations will be completely lost from genomes even in long-term evolution, although some processes, such as epistasis, may weaken the on-average relationship over time, and functionally important changes may be lost in a sea of inconsequential genotypic change. Convergence, when it occurs, is helpful to make such genotype-phenotype connections because repeated genotypic changes in the same phenotypic context are much less likely to occur by chance than single changes. Nevertheless, genotype-phenotype associations identified necessarily occur between branches on the phylogenetic tree on which multiple phenotypic changes may have occurred, particularly for complex phenotypes. Thus, we agree that even if truly convergent proteins have been identified, the link between the convergence and a particular phenotype should be evaluated with further biological analysis. In the section from L305 in the previous version of the manuscript, we analyzed molecular convergence associated with herbivory by identifying herbivory-associated branch pairs where convergent amino acid substitutions (nonsynonymous convergence) occurred with a high rate (ω_c). Our method detects “candidate” genes, so further support for causal relationships underlying genotype-phenotype associations should be obtained by consideration of what is known about the molecular underpinnings of a phenotype, and eventually experimental validation, as is the case in other *in-silico* analyses such as GWAS. Herbivory is a complex trait, and we discuss genes/proteins with convergent associations in relation to the molecular phenotypes that most likely constitute herbivory, such as bile acid and DNase properties. We show the amino acids encoded by the convergently occurred genotypes and their locations in protein structures in new Fig. S9 to clarify the question of which genes are involved. We believe that particularly promising candidate convergent genes with known molecular functions associated with herbivory include SLC51A, CYP7A1, and DNase I, and these are provided in the manuscript. Similar approaches have been adopted in many studies, including those cited by the “PhyloG2P” review discussed below. To further clarify these points, we made a few adjustments as follows.

Change: Identification of molecular convergence associated with a particular phenotype. Convergence metrics are often used to search for genes with substitutions that are repeatedly associated with

phenotypes of interest, indicating that such genes may often underlay similar phenotypic changes. we next examined whether ω_c might be used to discover candidate genes underlying phenotypic convergence. Here, we analyzed a pair of herbivorous animal lineages as an example of a search for genes associated with a complex trait (i.e., herbivory): ruminants (the stem branch of the clade including cattle [*Bos taurus*] and red sheep [*Ovis aries*]) versus rabbits (the terminal branch connected to *Oryctolagus cuniculus*). Using minimum thresholds for the number of convergent amino acid substitutions ($O_c^N \geq 3.0$) and protein convergence rate ($\omega_c \geq 3.0$), we obtained 352 candidate gene branch pairs corresponding to the above pair of lineages in a genome-scale analysis of the 21 vertebrates (Table S5). By mapping the positions of substitutions onto known conformations of homologous proteins, we identified particularly compelling cases of likely adaptive convergence that generated genotypes linked to particular phenotypes (Fig. S9). Examples included olfactory receptors in which convergent substitutions are located in the interior of the receptor barrel (ODORANT RECEPTOR 7A [OR7A], Olfactory Receptor Family 2 Subfamily M Member 2 [OR2M2], and OR1B1), where substitutions may change ligand preference associated with herbivorous behavior.

Similarly, the barrel-like structure of some solute carriers harbored convergent substitutions in their interior sides (Solute Carrier Family 5 Member 12 [SLC5A12], SLC51A, SLC22A, and SLC44A1), suggesting their involvement in the uptake or transport of plant-derived compounds. Among these, SLC51A (also known as Organic solute transporter α [OST α]) may be a particularly attractive candidate. This protein plays a major role in bile acid absorption and, hence, in dietary lipid absorption (Ballatori et al., 2005). The convergence in SLC51A may be coupled with another convergent event detected in CYP7A1, a cytochrome P450 protein known to serve as a critical regulatory enzyme of bile acid biosynthesis (Chiang and Ferrell, 2020). CYP7A1 harbored two convergent substitutions in its substrate-binding sites (Fig. S9). While most herbivores secrete bile acids mainly in a glycine-conjugated form, ruminant bile is mostly in the form of taurine-conjugated bile acids, which remain soluble in highly acidic conditions (Noble, 1981). The predominance of taurine-conjugated forms is also observed in rabbits, depending on species and developmental stage (Hagey et al., 1998). Thus, convergence in these proteins may be related to such nutritional physiology characteristics driven by herbivory.

Other examples of detected convergence included two convergent substitutions in the DNA-binding sites of a member of the zinc-finger protein family, which functions as a transcriptional regulator (Patel et al., 2018) (Fig. S9). Convergence in the substrate-binding sites of pancreatic elastase (Mulchande et al., 2007) and pancreatic DNase I (Weston et al., 1992) may be related to their specialized digestion (Fig. S9). In DNase I, amino acid sites exposed on the surface of protein structures displayed additional convergent substitutions that change the charge of their target amino acid residues (E124K, G172D, and H208N), possibly resulting in convergent changes in the biochemical properties of the protein, such as optimal pH, resistance to proteolysis, and posttranslational modifications. Consistent with this idea, bovine and rabbit DNase I proteins are known to be more resistant to degradation by pepsin than their homologs in other animals (Fujihara et al., 2012). Furthermore, E124K was shown to be important for the phosphorylation of bovine DNase I (Nishikawa et al., 1997). Other convergent substitutions will be promising candidates for future characterization. Taken together, these results show how our approach can detect genetic changes (e.g., molecular convergence in SLC51A, CYP7A1, and DNase I) associated with a phenotype (i.e., specialized digestion necessary for herbivory) on the macroevolutionary scale.

Comment 1.2.1. Even more worrying, the next section of the manuscript then describes how to look for convergence without a pre-specified phenotype, which will “provide a basis for understanding overlooked phenotypes”. But which ones? How would we identify these phenotypes? Especially on internal branches of species trees, I don’t have the foggiest idea how these signals would lead to an increased understanding of genotype-phenotype associations.

Response: We are sorry that we worried the reviewer, but even if internal branches are involved in a detected convergence event, many, if not all, extant species of the clades

involved could provide opportunities for the phenotypic association, mechanistic assessment, and experimental validation *in vivo*. This can be seen from the simple expectation that most species in the clade will tend to retain the convergent genotypes identified, and most species in the clade will tend to retain the phenotypic change that drove the molecular convergence. As an example of how we would identify associated phenotypes in a pair of species, we detected molecular convergence in liver-specific DHDH in the branches connected to *Xenopus* and *Astyanax*, which leads to a hypothesis that the two taxa share a novel detoxification ability in the liver. Although beyond the scope of this work, this hypothesis could be tested experimentally in a follow-up study. To further emphasize this point, we made the following changes.

Change: DHDH has broad substrate specificity for carbonyl compounds. This protein oxidizes *trans*-cyclohexanediol, *trans*-dihydrodiols of aromatic hydrocarbons, and monosaccharides including D-xylose, while it reduces dicarbonyl compounds, aldehydes, and ketones (Sato et al., 1994). Its active site is predominantly formed by hydrophobic residues, suggesting their role in catabolizing aromatic hydrocarbons (Carbone et al., 2008b, 2008a). Notably, the convergent substitutions in the substrate-binding pocket tended to increase amino acid hydrophobicity (Fig. S10B), suggesting that the remodeling of the active site may have led to the acquisition of new substrates, and hence a novel detoxification ability, in *Xenopus* and *Astyanax*.

In summary, ω_C was not only robust against phylogenetic errors, outperforming other methods in simulation and empirical data, but also allowed us to discover plausible adaptive convergence from a genome-scale dataset without a pre-existing hypothesis. The genotypes detected by molecular convergence analysis provide opportunities for the phenotypic association, mechanistic assessment, and experimental validation *in vivo*. This holds even if internal branches are involved in a detected convergence event, because most species in the clade will tend to retain the convergent genotypes identified, and most species in the clade will tend to retain the phenotypic change that drove the molecular convergence. Therefore, molecular convergence events revealed by our exploratory analysis provide a basis for understanding overlooked phenotypes that are in common among species in clades descended from branches where the convergent events occurred.

Comment 1.2.2. Putting aside the question of whether the method can help to identify such associations, if the authors really think this is a good use of their method, I think there are two key references missing: Smith et al. (2020, TREE) on the “PhyloG2P” approach, and Pease et al. (2016, PLoS Biology) on the “phyloGWAS” approach.

Response: We thank the reviewer for this suggestion. The pieces of literature were cited and discussed in the new version of the manuscript as below.

Change: Because of its improved accuracy, ω_C should further drive macroevolutionary analyses where uncorrected measures have been used to identify responsible genotypes for particular phenotypes in a way similar to genome-wide association studies (GWASs). It is noteworthy that there are more direct extensions of the GWAS approaches to analyze among-species variations. Those methods, including PhyloGWAS (Pease et al., 2016), can be applied to closely related species to detect convergent selection on ancestral variation or through introgression (Smith et al., 2020). Although those methods are powerful, the applicability to distantly related species is limited. In our method, as in alleles identified by GWASs or the above-mentioned comparable approaches (or genes in gene-level association tests (Wang et al., 2021)), genes with excess convergence serve as clues to study macroevolutionary traits for which the molecular basis is unknown (Fig. 5).

Comment 1.3.0. Convergence in duplicated genes. While I do see how a search for convergent evolution between gene duplicates would be very useful, I had a bit of trouble figuring out how this was actually done (there was very little description). For instance, are paralogs only in different species compared, or all pairs of paralogous branches? It seems like both might be informative, but about different biological processes. Similarly, why require two duplication events to separate paralogs ("DD")? It seems biologically more plausible that duplicates separated by only one duplication event (plus one speciation) would be more likely to converge, for the very same reasons that epistasis is invoked to explain decreasing convergence through time.

Response: In response to this comment, we clarified the design of our analysis in a new Method section and suggested a future analysis as below.

Change: Analysis of gene duplications. Gene duplications on gene trees were inferred by a species-overlap method, as explained above. Branches following gene duplication events are annotated as D branches. Note that, if one copy after the gene duplication is not included in the dataset due to poor gene annotation or other reasons (see Table S10 for gene completeness), the gene duplication node is lost and the branch that should have been classified as a D branch is combined with its parent branch. Such a bias would cause contamination from the D branch to the S branch, but the effect would be negligible because S branches are much more numerous than D branches and the opposite does not occur. Pairs of branches following two independently occurred gene duplications were extracted as DD branch pairs. In the genome-scale analysis, DD branch pairs may be connected to different species, or to the same species if successive duplications happen in the lineage. Nevertheless, paralogous gene lineages were compared in all cases. While the analysis of DD pairs was designed to characterize convergent gene duplications, one may wish to analyze the convergence of two copies generated by single gene duplication. In such an analysis, each copy must undergo speciation as soon as possible thereafter, since it is impossible to analyze convergence immediately after a duplication, where the branches are sisters to each other. Therefore, such an analysis will be best performed with a more densely taxon-sampled dataset to minimize the signal loss due to unanalyzable branches after duplications.

Comment 1.3.1. Also, how do you assign convergent substitutions to gene trees with missing genes? While this problem must also occur for single-copy genes, I imagine it must be much more common for trees with duplicates in them. And for trees with losses, it is not clear to me how branches can be compared fairly. Clearer explanations for the use of paralogs—and the patterns found (especially a bit more on the patterns described on lines 277-281)—would be helpful.

Response: In this study, we used gene trees throughout the manuscript, and as with all studies, are limited to the data and information available. Therefore, lost genes were not included in the analysis. The species tree was used in phylogeny reconciliation to improve gene tree topology, but this technique is compatible with gene loss and duplication. In response to this and the above comment, we added a new Method section.

Changes: See our response to Comment 1.3.0.

Comment 1.4.0. “Neutral” evolution. I admit this is a bit of a bugbear for me, but the use (and mis-use) of this term throughout the paper was both misleading and incorrect. Neutral evolution does not mean “no selection” or “no constraint”—it means the absence of positive selection (and the absence of balancing selection and weakly deleterious substitutions). While I know that there is a portion of the phylogenetic literature that says $\omega=1$ is evidence of “neutral evolution,” this is not correct. There is nothing about the neutral theory or neutral evolution that rejects a role for constraint, and this mis-use should really not be condoned. Even more importantly for the clarity of the arguments being made here: the authors only require the correct usage of neutral. In other words, the neutral expectation of ω_C “is” 1, because it is a ratio of convergent substitutions of each type. But this expectation does not require that $\omega=1$, even though this is what is said (line 159). In fact, it seems that the neutral simulations were actually done with $\omega=0.5$ (if I’m reading Table S2 correctly), which is perfectly sensible. (It seems like ω_C is mostly independent of the values of ω .) The conflation of these two uses of “neutral expectations” (one of which is not correct) simply made the manuscript harder to understand.

Response: We thank the reviewer for this comment, and we agree that despite potential misusage in the literature, it is important to clarify the difference between the situations where evolution is unconstrained and cases where evolution is constrained but the changes that occur are largely neutral (i.e., the Neutral Theory). In response to this comment, we made adjustments regarding the discussion concerning neutrality throughout the manuscript, as follows.

Change:

L95: A widely used framework for understanding how functionally constrained proteins evolve compared to **completely unconstrained** expectations is to contrast rates of nonsynonymous and synonymous substitutions, **using measures such as** dN and dS , or K_a and K_s , respectively (Yang, 2006).

L109: One of the most commonly accepted measures of the rate of protein evolution compared to **completely unconstrainedneutral** expectations is the ratio between nonsynonymous and synonymous substitution rates, denoted as dN and dS , or K_a and K_s , respectively (Yang, 2006).

L118: Using this ratio, important biological fluctuations, such as among-site rate heterogeneity and codon equilibrium frequencies, are taken into account (for details, see Supplementary Text 3 and Methods). **Note that in neutrally evolving genes, the theoretical expectation of ω_C is 1.0, even if ω in the underlying codon substitution matrix is not 1.0 (usually lower).** Similar to previously proposed convergence metrics (Castoe et al., 2009; Goldstein et al., 2015; Zou and Zhang, 2015a), ω_C is calculated from substitutions at multiple codon sites across protein-coding sequences.

L133: Conventionally, observed levels of convergent amino acid substitutions have been contrasted either to the amount of convergence expected under a **substitutionneutral model-with-no-constraint** (R (Zou and Zhang, 2015a)) or to other combinations of amino acid substitution patterns that are similarly affected by site-specific constraint (i.e., double divergence; C/D (Castoe et al., 2009; Goldstein et al., 2015)) (Table S1; Supplementary Text 4).

L308: Furthermore, we examined the synonymous convergence rate (dS_C), which is not expected to be greater than the ~~theoretical neutral~~ expectation in the adaptive convergence, and established that only ω_C satisfies such an assumption (Fig. 3A).

L1761: According to the value of ω , the mode of protein evolution can be categorized into purifying selection ($\omega < 1$), ~~neutral evolution without constraint~~ ($\omega = 1$), and adaptive evolution ($\omega > 1$).

Comment 1.4.1. Similarly, it is not clear to me what the line for “Theoretically Neutral” is pointing to in the panels of Figure 1D with dN_C and dS_C in them. Why should these two metrics have an expectation of 1?

Response: Both dS_C and dN_C are the observed number divided by the expected number. If the theoretically expected number derived from the codon substitution model fully explains the observed number, then both dS_C and dN_C values would be 1.0. For figure labels, “Theoretical expectation” will be less confusing. In response to this comment, we replaced the “neutral expectation” labels in all figures with “theoretical expectation” and explained it in the Methods.

Change in Methods: Both dN_C and dS_C are the observed number divided by the expected number. If the theoretically expected number derived from the codon substitution model fully explains the observed number, then both dN_C and dS_C values would be 1.0.

Change in Figures: See the following figures in the new manuscript: Fig. 1D,E, Fig. 3A, Fig. 5, Fig. S4, Fig. S5I, Fig. S8B, Fig. S14A, and Fig. S15.

Comment 1.4.2. The citation to Bustamante et al. 2005 (line 780) was also confusing, as the McDonald-Kreitman test isn’t a test for purifying selection. The results in that paper simply say that there are segregating weakly deleterious polymorphisms in humans.

Response: In response to this comment, we removed the citation.

Change: Under purifying selection, which is the default evolutionary mode of many proteins (~~Bustamante et al., 2005~~), the rate of synonymous substitutions is faster than that of nonsynonymous substitutions.

Comment 1.4.3. Finally, “nearly neutral” and “essentially nonfunctional” (line 75) are not contrasting ideas—the former is about the fitness effect of substituting one allele for another, and the latter describes a change in molecular function.

Response: We agree and did not mean to imply that these were linked as contrasting ideas, so in response to this comment we revised the manuscript as follows.

Change: Genome sequences are becoming more available for diverse lineages from the entire tree of life (Lewin et al., 2022), making it possible to explore macroevolutionary genotype-phenotype associations on large scales. However, because many molecular changes are nearly neutral (i.e., almost no effect on fitness) and/or essentially nonfunctional in nature (Ohta, 2002), false-positive convergence in the form of stochastic, nonadaptive, convergent events is particularly problematic when conducting a genome-scale search.

Comment 1.5.0. Minor comments. -dN and dS are not rates, they are distances.

Response: We agree that dN and dS are sometimes viewed as numbers or distances, but it is always implicit that they are numbers that occurred on a branch, that is, over a specific period of time. The notation of HyPhy (<http://hyphy.org/>) and PAML (<https://academic.oup.com/mbe/article/24/8/1586/1103731>), which are most frequently used to calculate dN and dS in the field of molecular phylogenetics, and in which dN and dS are referred to as rates, is therefore also correct. We will refer to them as rates in this manuscript to be consistent with this notation.

Comment 1.5.1. -I am not sure what the section on “Extracting a high-confidence set of convergent lineages” (starting line 287) shows, exactly. That there are outliers on any pair of branches?

Response: This section analyzes which convergence metrics yield plausible sets of convergent branch combinations. The top 1% branch combinations in each metric are analyzed so that different convergence metrics can be compared without introducing arbitrary thresholds in each metric. To further clarify this point, we changed the manuscript as follows.

Change: ω_c probes a high-confidence set of convergently evolved proteins. Discovering adaptive molecular convergence in genome-scale datasets, which may be translated into genotype-phenotype associations, has been challenging since it is a rare phenomenon and false positives are high (Foote et al., 2015; Thomas and Hahn, 2015; Zou and Zhang, 2015b). To examine whether the application of ω_c can generate plausible hypotheses of adaptive molecular convergence, we analyzed the 21 vertebrate genomes (Fig. S7A). We first extracted the branch pairs with the top 1% of C/D , dN_c , or ω_c values with a cutoff for a minimum of three nonsynonymous and synonymous convergence ($O_c^N \geq 3.0$ and $O_c^S \geq 3.0$) (Fig. 3A). The top 1% threshold allows different convergence metrics to be compared without introducing arbitrary thresholds in each metric. The overlap between each set of branch pairs was moderate, with 1,348 branch pairs satisfying all three criteria out of 5,659 pairs with the top 1% ω_c values.

Comment 1.5.2. -Could the authors comment on the relationship of the metrics they use to the one proposed in Burskaia et al. (2020, GBE)?

Response: In response to this comment, we clarified the difference between ω_c and their P in Methods as follows.

Change: There is another metric to analyze molecular convergence using both nonsynonymous and synonymous substitutions (Burskaia et al., 2020). This metric, called P , contrasts the proportion of nonsynonymous convergence at nondegenerative nucleotide sites (their dN_p) and the proportion of synonymous convergence at four-fold degenerate nucleotide sites (their dS_p) in phylogenetic quartets. ω_c is distinct from P in many aspects, including the use of complete phylogenetic trees rather than decomposed quartets, the use of all codon sites regardless of their degree of codon degeneration, and the use of expected values based on a codon substitution model rather than the proportion of convergent substitutions.

Comment 1.5.3. -Figure S3C seems to show quite a bit of sensitivity of omega_c to misspecification of the codon model. Maybe this should be mentioned in the main text?

Response: In response to this comment, we revised the manuscript as follows.

Change: Further simulations supported the robustness of C against the rate of protein evolution, model misspecification, tree size, and protein size (Fig. S3B–F). Still, care must be taken when using simple codon substitution models, such as MG and GY models (Supplementary Text 5). Furthermore, C showed low false-positive rates in sister branches that serve as a control for the focal branch pairs (Foote et al., 2015) (Fig. S3G).

Comment 1.5.4. - It was not clear to me that the expression changes detected are adaptive. What if there is a lot of overlap with testis expression because this pattern accumulates at high (neutral) rates?

Response: In response to this comment, we performed a set of chi-square tests for the enrichment of joint convergence in brach pairs where both gene lineages express in a particular tissue. Indeed, testis showed the largest number of highest-expressed genes, but such among-tissue heterogeneity is controlled for in these tests because chi-square tests of independence were performed within those genes highest-expressed in a particular tissue. Testis showed significant enrichment of joint convergence, with the highest effect size among six tissues. While this result clearly suggests the uniqueness of testis, it is still difficult to distinguish the effects of adaptation and neutral evolution, as the reviewer mentioned. To better describe it, we revised the manuscript as follows.

Change: Convergence of testis-specific genes was most frequently observed (19/33 orthogroups) and significantly enriched, with the effect size highest among six tissues (Table S7). The mechanism by which the testis serves as a major place for functional evolution of duplicated genes has been explained by several factors, including the ease with which expression is acquired in spermatogenic cells (Kaessmann, 2010; Kleene, 2005). This phenomenon is called the out-of-the-testis hypothesis, and our results suggest that predictable protein evolution may be enriched in this evolutionary pathway. While adaptive evolution may explain this evolutionary scenario, it is possible that partially relaxed constraints may also be involved in protein convergence, particularly at protein sites that were so constrained that almost no amino acid substitutions occurred before relaxation.

Table S7. Tissue-wise enrichment of joint expression–protein convergence.

Highest expression	Number of branch pairs	Number of protein convergence	Number of expression convergence	Observed number of joint convergence	Expected number of joint convergence	Chi-square	P value
Brain	3,915,347	11,392	909	3	2.644804662	0	1
Heart	1,521,046	4,352	207	0	0.5922661116	0.0144 168719 2	0.90442 74759
Kidney	867,802	3,641	220	1	0.9230446577	0	1
Liver	2,494,826	8,647	429	6	1.486902493	10.870 74949	0.00097 6949859 5
Ovary	1,780,434	4,293	312	4	0.7522974735	10.061 76973	0.00151 3777455
Testis	4,715,493	11,200	840	19	1.99512543	136.88 76803	1.28E- 31

Comment 1.5.5. -Figure 3 was not very informative. Maybe some more detail on the expression method/patterns, and less on protein structures?

Response: To illustrate how convergent expression shifts are modeled, we included new Fig. 3B. This change was possible without reducing the information of protein structure, which is crucial to illustrate how convergent amino acid substitutions can change protein functions, as described in detail in the main text.

Change in main text: Using this previously published dataset, we subjected curated gene expression levels (SVA-log-TMM-FPKM) to multi-optima phylogenetic Ornstein-Uhlenbeck (OU) modeling, in which expression evolution is inferred as regime shifts of estimated optimal expression levels (Khabbazian et al., 2016) (Fig. 3B).

Change in Figure:

Figure 3. Joint convergence of gene expression patterns and protein sequences. (A) Comparison of convergent branch pairs obtained by different methods in the vertebrate dataset. Branch pairs with $O_c^N \geq 3.0$ and $O_c^E \geq 3.0$ were analyzed. The Venn diagram on the left shows the extent of overlap between the top 1% convergent branch pairs. The violin plot in the middle shows the lower bootstrap support of the parental branches of the convergent branch pairs. The boxplot on the right compares the rate of synonymous convergence (dS_c). The stochastic equality of data was tested by a two-sided Brunner–Munzel test (Brunner and Munzel, 2000). (B) A schematic illustration of convergent expression evolution modeled with multi-optima Ornstein-Uhlenbeck process. (C) Venn diagrams showing the extent of overlap between protein and

expression convergence. Circles represent the sets of branch pairs. Shifts in tissue-specific expression regime were identified with the thresholds of expression levels (the maximum fitted SVA-log-TMM-FPKM among tissues (Fukushima and Pollock, 2020)) and tissue specificity (Yanai's τ (Yanai et al., 2005)). (D–G) Examples of the likely adaptive joint convergence. Aldo-keto reductase family 1 (AKR1, D), Nudix hydrolase 16 like 1 (NUDT16L1, E), Myeloid associated differentiation marker (MYADM, F), and Dihydrodiol dehydrogenase (DHDH, G) are shown (see Fig. S9A for complete trees). Node colors in the trees indicate inferred branching events of speciation (blue) and gene duplication (red). The heatmap shows expression levels observed in extant species. The silhouettes signify the species (see Fig. S7A) that carries the gene, and the clades involved in the joint convergence are indicated with an enlarged size. The colors of branches and animal silhouettes indicate expression regimes. Among-organ expression patterns are shown as a pie chart for each regime. Branches involved in joint convergence are highlighted with thick lines, connected by the color of the expression regime, and annotated with convergence metrics. Localization of convergent and divergent substitutions on the protein structure is shown along with a close-up view of functionally important sites. The surface representation of each protein is overlaid with a cartoon representation. Convergent and divergent amino acid loci shown in Fig. S9 are highlighted in red and blue, respectively. Substrates and their analogs are shown as green sticks. Side chains forming the substrate-binding site are also shown as sticks. Note that these are the side chains in the protein from databases, so amino acid substitutions in the convergent lineages may result in distinct structures and arrangements. Site numbers correspond to those in the PDB entry or the AlphaFold structure (from D to G: 1Q13, 5W6X, AF-Q6DFR5-F1-model_v2, and 2O48). The silhouettes of *Astyanax mexicanus* and *Oreochromis niloticus* are licensed under CC BY-NC-SA 3.0 (<https://creativecommons.org/licenses/by-nc-sa/3.0/>) by Milton Tan (reproduced with permission), and those of *Anolis carolinensis* (by Sarah Werning), *Ornithorhynchus anatinus* (by Sarah Werning), and *Rattus norvegicus* (by Rebecca Groom; with modification) are licensed under CC BY 3.0 (<https://creativecommons.org/licenses/by/3.0/>).

Comment 1.5.6. -Is there a formula for number of independent branches in different sized trees? I'm just wondering whether there's an easy way to get the number laid out starting on line 446, as it seems like a very helpful calculation.

Response: The number of independent branches may differ due to tree topology even if the number of OTUs is the same, and we don't know of a simple formula. To obtain the numbers, we gave CSUBST manually generated trees and let it count the branch combinations, which solved the problem well enough that we did not spend further time on it.

Comment 1.5.7. -The "branch-and-bound" algorithm described seems to be in fact be a greedy algorithm. I think the same approach is often taken in trying to detect pairwise epistasis in mapping studies.

Response: Our method is similar to the limitless arity multiple-testing procedure (LAMP, <https://www.pnas.org/doi/10.1073/pnas.1302233110>), which uses a branch-and-bound algorithm for the detection of higher-order transcription factor combinations in gene regulation, but as the reviewer points out, it can also be viewed as a greedy algorithm. In response to this comment, we revised the manuscript as follows.

Change: To overcome this limitation, we developed an efficient branch-and-bound algorithm (Land and Doig, 1960) that progressively searches for higher-order branch combinations (Fig. 4A and Fig. S12A), similar to a method used for the detection of higher-order transcription factor combinations in gene regulation (Terada et al., 2013). This method can also be considered a type of greedy algorithm because it determines the search range for higher-order branch combinations based on the convergence rate of lower-

order branch combinations. For the performance evaluation, we used the PEPC tree (Fig. 4B) because it has repeated adaptive convergence for its use in C₄ photosynthesis (Fig. 1E).

Reviewer #2

Comment 2.0.0. In this manuscript, Fukushima and Pollock develop a new method to detect protein-coding genes with an excess of convergent amino acid substitutions, that corrects for the biases and false positives that have plagued this field for a while. Their approach, implemented in the software package CSUBST, appears to be robust and performs well in the tests reported in this manuscript, making it highly likely this will be adopted as a standard method in the community of evolutionary biologists who study convergent evolution. I would further note that the code associated with the manuscript is notably well organized, easy to install, and simple to work with, with multiple example datasets and a well organized wiki/help section. Nonetheless, the manuscript would be significantly improved by some revisions, which I outline here. Of particular importance are the technical issues in point 4 below.

Response: We thank the reviewer for their helpful comments and appreciation of the technical advance of this work.

Comment 2.1.0. As the authors mention in their discussion (503-511), approaches such as CSUBST rely on identifying genes with more convergent substitutions than expected under some kind of neutral model; this approach can have low power when there are only one or a few convergent substitutions in a protein. While CSUBST is innovative in using convergent synonymous substitutions to normalize the excess over expectations in nonsynonymous substitutions, there is an alternate line of research, perhaps best exemplified by Marcovitz et al 2019 (PMC6800341), that instead considers the full collection of convergence substitutions and asks questions about functional enrichment of the genes affected. I would be very interested to see if there is a way that these ideas could be merged with the authors' method. For example, it seems like it could be possible to aggregate by, e.g., some kind of functional class instead of gene, to gain significantly more power to detect cases of adaptive convergence. This could be conceptually similar to methods like INSIGHT (Gronau et al 2013, <https://doi.org/10.1093/molbev/mst019>), which aggregate "genomically coherent elements" for tests of selection to increase power when the substitutions of interest are rare. I realize that while fully implementing this kind of extension could be beyond the scope of the current manuscript, a more extensive discussion of limitations and possible extensions of the CSUBST method would help contextualize the current work and its potential and future impact.

Response: In response to this comment, we revised the Discussion as follows.

Change: If many amino acid sites and/or many separate lineages are involved, true convergence is, in general, more easily detected. However, it should be noted that errors due to splicing variants may not be completely eliminated in higher-order branch combinations (Supplementary Text 10). Alternatively, it may be possible to detect weak convergent signals by concatenating genes in similar functional categories (Marcovitz et al., 2019). A limitation of such an approach is that groups of genes containing lineage-specific duplications or losses cannot be analyzed.

Comment 2.2.0. Related to this point, a challenge for many studies seeking to identify convergent amino acid substitutions is that there is the possibility that this is simply not a particularly common phenomenon. While a few key examples are repeatedly discussed (PEPC in C4 photosynthesis, prestin in echolocating mammals, ATPalpha1 in glycoside resistant insects, etc), an exciting possibility of the new method the authors propose is the potential to do exploratory analysis in the absence of known candidates, as the authors discuss in the section starting on line 344 and again the discussion, as well as showing in Fig 3 and Fig 5. However, it is not particularly clear from the results the authors' present how well this works. While I understand the motivation for focusing on the small number of genes that show overlapping patterns of convergence between gene expression and amino acid datasets, the results of this analysis seem more relevant to questions of the role of gene duplication in protein evolution than the kind of genotype-phenotype link the authors emphasize in e.g. Fig 5. Therefore, I think that a few additional analyses in this section would be particularly valuable. i. First, the authors identify 53,805 candidate branch pairs in their exploratory analysis. This would average to about 100 cases of excess convergence for a pair of species, as the authors discuss on lines 351-356. It is notable, to me, that the number of branch pairs with excess convergence in the herbivore analysis is substantially higher than this average, at 352. This raises the question of what the distribution of these number actually are. Do species pairs with well-known convergent phenotypes tend to have higher-than-average numbers of genes with excess convergence? The general conclusion in many previous studies has been that this is not the case, but given that previous methods were very sensitive to false positives this could swamp real signal.

Response: A claim for the genome-scale elevation of convergence rates requires particularly thorough characterization given the extensive discussion in the past decade, and therefore it is beyond our scope. However, the reason the number of convergently evolved genes is higher in herbivores may come from several factors that are non-adaptive per se. This is now explained as follows with new Fig. S10 and Supplementary Text 9. The number of combinations discussed in the main text has also been revised since a close examination of the data revealed that in some cases, even if the species branch pair is not independent, the corresponding gene branch pairs can be phylogenetically independent, and such cases have not been taken into account in the previous version of the manuscript.

Change in main text:

For example, because there are 861 ~~independent~~ branch pairs in the species tree, on average 62.5 cases of protein convergence will be obtained in our genome-scale dataset for any particular analysis of two groups

of species, although the numbers of analyzable branch pairs and hence of detected convergence depend on several factors (Supplementary Text 9 and Fig. S10).

Change in Supplementary Text:

Supplementary Text 9. Factors affecting the number of branch combinations in genome-scale analysis.

The number of gene branch combinations mapped to a species branch combination depends on a variety of factors, and therefore the number of detected convergence per species branch pair can vary non-adaptively. Overall, terminal branches in the species tree tended to involve larger numbers of gene branch combinations than internal branches (Fig. S10A). This is likely because the more terminal the species branches, the more gene branches for comparison due to the accumulation of branches generated by historical gene duplications. Another factor that may explain the differences between internal and terminal branches is branch lengths in the species tree, which should be correlated with the number of gene duplications under a relatively constant gene duplication rate. Indeed, the product of species branch lengths showed a moderate correlation with the number of gene branch combinations (Spearman's $\rho = 0.283$, Fig. S10A). Inheriting this heterogeneity, the number of convergent branch pairs also varies greatly among species branch pairs (Fig. S10B), with Spearman's correlation coefficient as high as 0.745 (Fig. S10C). The terminal branch connected to *Danio rerio* showed an unusually high number of gene branch combinations (diagonal elements in Fig. S10A,B). This feature may partly be explained by the number of annotated genes in this species, which was the highest among analyzed genomes (Fig. S10D). In large orthogroups dominated by *Danio* genes, a large number of branch pairs should be generated for the comparison of two *Danio* gene lineages.

Change in Supplementary Figure:

A All branch combinations ($\omega_c \geq 0.0$ and $O_c^N \geq 0.0$)

Figure S10. Number of branch combinations in the animal genome analysis. (A) Number of gene branch pairs per species branch pair. Internal branch names are indicated in the species tree. No convergence threshold is applied (i.e., $\omega_c \geq 0.0$ and $O_c^N \geq 0.0$). Note that even identical branches or branches in an ancestor-descendant or sister relationship in the species tree can be independent in gene trees if there is a preceding gene duplication event. Some animal silhouettes were obtained from PhyloPic (<http://phylopic.org>). The silhouettes of *Astyanax mexicanus* and *Oreochromis niloticus* are licensed under CC BY-NC-SA 3.0 (<https://creativecommons.org/licenses/by-nc-sa/3.0/>) by Milton Tan (reproduced with permission), and those of *Anolis carolinensis* (by Sarah Werning), *Ornithorhynchus anatinus* (by Sarah Werning), and *Rattus norvegicus* (by Rebecca Groom; with modification) are licensed under CC BY 3.0 (<https://creativecommons.org/licenses/by/3.0/>). (B) Number of convergent gene branch pairs per species branch pair under the threshold of $\omega_c \geq 3.0$ and $O_c^N \geq 3.0$. (C) Relationships between the numbers of all branch pairs and convergent branch pairs. (D) Number of analyzed genes per genome.

Fig. S10 (continued)

Comment 2.2.1. ii. Second, from the analysis of PEPC in C4 photosynthesis plants, it seems like a genome-wide, unbiased screen for say 3- or 4-way convergence in the mammalian tree (something that has not really been possible before CSUBST) could be extremely informative to identify proteins that are highly likely to be biological relevance for interesting phenotypes. This kind of analysis would go a long way towards demonstrating the clear potential of CSUBST for the kinds of exploratory analyses the authors pitch as one of the impactful areas of study enabled by their work.

Response: In response to this comment, we performed a genome-scale analysis of higher-order convergence considering up to 10 lineages. The results were explained in the new manuscript as follows.

Change in main text: In the higher-order C4 branch combinations, the detected convergence events were almost entirely nonsynonymous (OCN), while synonymous convergence (OCS) was negligible (Fig. 4D). As a result, the rate of synonymous convergence (dSC) quickly approached zero (Fig. 4D). Notably, the higher-order convergent substitutions were located at functionally important protein sites. In the convergent branch combinations with K=6, we identified three amino acid sites with a joint posterior probability of nonsynonymous convergence greater than 0.5: V627L, H665N, and A780S (Fig. S12B–D). The H665N substitution generates a putative N-glycosylation site that may be important for protein folding (Christin et al., 2007). The A780S substitution, for which the signature of positive selection had been detected previously (Besnard et al., 2009; Hermans and Westhoff, 1992; Poetsch et al., 1991), has been shown to change the enzyme kinetics related to the first committed step of C4 carbon fixation (Bläsing et al., 2000; DiMario and Cousins, 2019; Engelmann et al., 2002) and is therefore considered a diagnostic substitution of C4-type PEPC (Besnard et al., 2009; Christin et al., 2007). The third substitution, C627L, might be a good focus for future experimentation. **Application of the heuristic search to the 21 animal genomes revealed that while likely adaptive higher-order convergence could be detected, false detections arising from inconsistently represented splicing variants should be carefully avoided when performing genome-scale analyses (Supplementary Text 10, Fig. S13, and Table S9). Nevertheless, these results demonstrate that higher-order analysis can substantially increase the signal-to-noise ratio in convergence analysis when there is repeated selective pressure to evolve similar biochemical functions.**

Change in Supplementary Text: Supplementary Text 10. Higher-order convergence in the 21 animal genomes. To further characterize the heuristic search of highly-repetitive convergence, we again analyzed the 21 animal genomes. The same threshold as in the analysis of PEPC (C5.0 and OCN2.0) was applied to search branch combinations up to K=10 (i.e., convergence among 10 branches). Up to K=3, the numbers of convergent branch combinations were two orders of magnitude less than the numbers of analyzed combinations, but thereafter, the difference was drastically reduced, indicating an efficient search of branch combination space (Fig. S13A and Table S9).

At K=10, only two out of 16,724 orthogroups were detected to contain convergent combinations. Upon closer examination, one of them (OG0000136, encoding Glutamate receptors and containing 742 out of 746 detected combinations at K=10) was found to be a likely artifact due to different splicing variants being inconsistently included in the representative gene set for each species (Fig. S13). In the animal genome analysis, we selected the longest transcript among splicing variants according to common practice (Fukushima and Pollock, 2020), and this operation seems to create the artifacts. A characteristic feature of this artifact is that many combinatorial substitutions are concentrated to a narrow window of the protein sequence (Fig. S13B). Protein convergence at K=2

shown in Fig. S11A did not show such a feature, and therefore this problem may be pronounced particularly when analyzing higher-order convergence. We expect synonymous convergence to cancel out the false signal in many cases, but in the cases where synonymous substitutions did not happen or are largely lost, the artifacts are not completely excluded from the results of genome-scale analyses.

The other detected orthogroup at $K=10$ (OG0000062, encoding Protocadherin beta) did not show the signature of false convergence due to inconsistently represented alternative transcripts (Fig. S13C). Only two lineages were involved in the four sets of 10 detected branches: pigs and the lineage connected to mice and rats (Fig. S13D). At such a high-order convergence, no synonymous convergence was not detected at all, so C diverged to infinity in four detected branch combinations. Although alternative mechanisms such as gene conversion may be involved, this orthogroup may represent a case of biologically generated highly-repetitive convergence, with a possibility of a highly coevolving pair of amino acid sites in a unit of the extracellular cadherin repeats (Fig. S13E). In the two amino acid sites, it appears that the same substitutions occurred outside of the 10 lineages, but they were not detected with the threshold we used (Fig. S13D). Although only 21 species were included in this genome-scale analysis, a larger set of genomes will enable the detection of higher-order molecular convergence that correlates well with phenotypes, as in the case of PEPC.

Change in Supplementary Table:

Table S9. Orthogroups and branch combinations detected in the genome-scale analysis of higher-order convergence.

K	Number of convergent branch combinations	Number of all examined branch combinations	Number of orthogroups containing convergent branch combinations	Number of orthogroups containing all examined branch combinations
2	70071	20150538	10513	16724
3	2276	320325	361	4230
4	1712	3324	64	112
5	2760	3871	25	29
6	4016	6040	9	10
7	4487	7810	5	6
8	3619	7595	5	5
9	2027	5310	3	3
10	746	2581	2	2

Change in Supplementary Figure:

Figure S13. Analysis of highly repetitive convergence in 21 animal genomes. (A) Numbers of orthogroups and branch combinations in the higher-order analysis. (B) Falsely detected protein convergence in OG0000136 at $K = 10$. Combinatorial substitutions are clustered to a limited range of the protein sequence. To the right, all alternative transcripts in human genes annotated in Ensembl are shown. Alternative transcripts from multiple genes harbor the same set of variations, likely generating false convergence. (C) Protein convergence in OG0000062 at $K = 10$. No evidence was found for shared variations among alternative transcripts. (D) Convergent branch combinations in the OG0000062 tree at $K = 6$. (E) Positions of higher-order convergent substitutions in the structure of a protocadherin ectodomain (PDB ID: 6VG4) (Harrison et al., 2020).

Fig. S13 (continued)

Comment 2.3.0. A minor point about the CSUBST program. While the documentation for installation and basic usage is admirably clear, and the examples for advanced usage are nice to see, the program has a huge number of options that are not clear how to set or use, many of which I presume the user will rarely, if ever, want to modify. A short addition to the readme that describes the submodules (dataset, analyze, simulate, site) with a brief description of the options the user is most likely to want to change, would be extremely helpful.

Response: In response to this comment, we revised the README of the GitHub repo (<https://github.com/kfuku52/csubst>). The most relevant addition is as follows.

Change in GitHub site:

CSUBST is composed of several subcommands. `csubst -h` shows the list of subcommands, and the complete set of subcommand options are available from `csubst SUBCOMMAND -h` (e.g., `csubst analyze -h`). Many options are available, but those used by a typical user would be as follows. More advanced usage is available in CSUBST wiki.

- `csubst dataset` returns an out-of-the-box test datasets.
 - `--name`: Name of dataset. For a small test dataset, try `PGK` (vertebrate phosphoglycerate kinase genes).
- `csubst analyze` is the main function of CSUBST. This subcommand returns various files including a table for ω C, dNc, and dSc.
 - `--alignment_file`: PATH to input in-frame codon alignment.
 - `--rooted_tree_file`: PATH to input rooted tree. Tip labels should be consistent with `--alignment_file`.
 - `--genetic_code`: NCBI codon table ID. 1 = "Standard". See here for details.
 - `--iqtree_model`: Codon substitution model for ancestral state reconstruction. Base models of "MG", "GY", "ECMK07", and "ECMrest" are supported. Among-site rate heterogeneity and codon frequencies can be specified. See IQTREE's website for details.
 - `--threads`: The number of CPUs for parallel computations (e.g., 1 or 4).
 - `--foreground`: Optional. A text file to specify the foreground lineages. The file should contain two columns separated by a tab: 1st column for lineage IDs and 2nd for regex-compatible leaf names.
- `csubst site` maps combinatorial substitutions onto protein structure.
 - `--alignment_file`: PATH to input in-frame codon alignment.
 - `--rooted_tree_file`: PATH to input rooted tree. Tip labels should be consistent with `--alignment_file`.
 - `--genetic_code`: NCBI codon table ID. 1 = "Standard". See here for details.
 - `--iqtree_model`: Codon substitution model for ancestral state reconstruction. Base models of "MG", "GY", "ECMK07", and "ECMrest"

are supported. Among-site rate heterogeneity and codon frequencies can be specified. See IQTREE's website for details.

- `csubst simulate` generates a simulated sequence alignment under a convergent evolutionary scenario.
 - `--alignment_file`: PATH to input in-frame codon alignment.
 - `--rooted_tree_file`: PATH to input rooted tree. Tip labels should be consistent with `--alignment_file`.
 - `--genetic_code`: NCBI codon table ID. 1 = "Standard". See here for details.
 - `--iqtree_model`: Codon substitution model for ancestral state reconstruction. Base models of "MG", "GY", "ECMK07", and "ECMrest" are supported. Among-site rate heterogeneity and codon frequencies can be specified. See IQTREE's website for details.
 - `--foreground`: A text file to specify the foreground lineages. The file should contain two columns separated by a tab: 1st column for lineage IDs and 2nd for regex-compatible leaf names.

Comment 2.4.0. I also have some technical questions that should be addressed. i. The authors use a variety of $\omega[c]$ and $\text{observed}[N][c]$ thresholds to identify genes with excess convergence, and these thresholds vary between the higher-order branch-and-bound search and the exhaustive pairwise search. However, there is little/no discussion in the paper of why these thresholds were chosen (unless I missed it, in which case I apologize). I think this really needs to be clearly explained and justified. Is there some reason to believe that an $\omega[c]$ value of 3 is a particularly important cutoff? Is this what seems to work in practice, or is there a statistical justification for it? Are the same cutoffs likely to be universally useful regardless of tree size and species included, or is this something the user is likely going to have to optimize? Of particular concern is the fact that several of the key examples presented in Fig 1 / Table S4 (e.g., prestin with $O_C^N \sim 1.09$, ATPalpha1 with $O_C^N \sim 1.4$) don't seem to pass these same thresholds. The simulation data shows the advantages of $\omega[c]$ clearly, but it is less obvious, in practice, how one translates this to real data. The method seems fast enough that adding some kind of simulation or permutation based P-value may be of significant value and may need to be explored to address some of these issues.

Response: In response to this and other comments, we performed an analysis of sensitivity. Please see our response to Comment 1.1.0. As a small number of nonsynonymous convergence is challenging to detect, the sensitivity of this range ($1 < O_C^N < 2$) is low (36% in this simulation). Although simulation is useful in many aspects, we want to be cautious about obtaining P values from simulations with parameters that mimic real data, because ordinary simulations of molecular evolution (including ours) assume independence between sites, and the lack of epistasis will almost always lead to an underestimation of convergence rates (i.e., inflated false positives). Thermodynamic simulations of protein evolution could overcome this

problem, but it is not realistic to perform a large number of simulations, and also it is difficult to apply them to real data. Permutations were difficult too because it was not obvious how to randomize the posterior probabilities of substitutions that occur on phylogenetic branches to obtain a null distribution of convergence rates. We believe this is a worthy issue to address, but it is too challenging to be included as a part of this paper. However, we included a new paragraph in Discussion where thresholds were discussed. Please see our response to Comment 1.1.1.

Comment 2.4.1. ii. After digging into the results in Tables S5 and S6, I am a little bit confused about how large gene families with speciation and duplication events are being analyzed. For example, looking at OG0000039, there are four branch pairs with excess convergence. Both 155/202 and 155/209 are listed as 'SS', as in both branches are speciation events. However, it seems to me that 202 and 209 must represent different paralogs, right? So, I think you have convergence between paralog A[rabbit] and paralog A[ruminant], as well as convergence between paralog A[rabbit] and paralog B[ruminant]. Something similar seems to happen in OG0000059. I don't really understand in this case why all of these are being listed as 'SS' branching types. I think a supplemental figure or diagram would be very helpful here to clarify what is going on. Additionally, I think it would be extremely useful to have a way to filter to identify convergent events only between branches where the most recent common ancestor of the two convergent branches is a speciation event. **iii.** Related to the last point, I am in general somewhat confused as to how gene duplication is treated in this manuscript. Looking at Figure 1C, it seems for the simplified tree ((species 1 copy 1, species 2 copy 1),(species 1 copy 2, species 2 copy 2)) convergence between copy 1 in species 1 and copy 2 in species 1 would be treated as a 'SS' event, since the immediately preceding node in each case is a speciation event. I am not sure if this really makes sense? It would perhaps be more intuitive to classify branch pairs by whether their common ancestor is a speciation or a duplication event, especially as in the current framework it is not obvious at all what an 'SD' event means.

Response: The branch categories S and D refer to which of speciation and duplication the immediately preceding branch event was, and therefore the branching event of the most recent common ancestor of two deeply diverged branches can be a duplication even if both branches are labeled S. We used this categorization to analyze molecular evolution immediately after gene duplication. SD is a pair in which one branch is produced by a speciation and the other by a duplication. If branches are classified by branching events in the common ancestor, branches that have experienced a significant amount of time after the event will be analyzed. Distant gene lineages would inevitably contain multiple gene duplications between them, and this would make it difficult to detect the effect of the original event on the common ancestor, and therefore we classified branches with their immediately preceding events. In response to this and other comments, we clarified our gene duplication analysis. Please see our response to Comment 1.3.0.

Decision Letter, first revision:

5th August 2022

Dear Dr. Fukushima,

Thank you for submitting your revised manuscript "Detecting macroevolutionary genotype-phenotype associations using error-corrected rates of protein convergence" (NATECOLEVOL-220316177A). It has now been seen again by the original reviewers and their comments are below. The reviewers find that the paper has improved in revision, and therefore we'll be happy in principle to publish it in Nature Ecology & Evolution, pending minor revisions to satisfy the reviewers' final requests and to comply with our editorial and formatting guidelines.

[REDACTED]

Reviewer #1 (Remarks to the Author):

Re-review of "Detecting macroevolutionary genotype-phenotype associations using error-corrected rates of protein convergence"

The updated manuscript clarifies many of the issues raised in the first round of review, and I thank the authors for their many changes and updates. I still do not really agree with the focus on possible genotype-phenotype associations, but I at least agree that convergent evolution is one way to (indirectly) try to identify such associations.

Two remaining minor points that I don't feel are adequately addressed yet:

- "demonstrating that genotype-phenotype associations frequently occur on macroevolutionary scales" (lines 67-68). What does this mean? Are the authors trying to say something about the origins of traits, or our ability to detect them? What does convergent evolution have to do with the existence of such associations? Presumably there is an association between genotype and phenotype regardless of whether there is convergence.

33-The authors have clarified that they have tested for convergence among duplicated genes within the same species. What phenotypes can duplicated genes from the same species be associated with? Related to this, the authors have also clarified that when one duplicate is lost, the remaining branch is "contaminated" with substitutions from the neighboring branch (because they cannot be separated). But what happens when a single-copy gene is lost from just one species? The previous answer implies there are none of these, but I can't tell whether this is the case or whether there was some confusion about what I meant by "lost genes". Can the authors please clarify?

Reviewer #2 (Remarks to the Author):

The authors have added a number of useful analyses, figures, and text that have in my opinion done an excellent job of addressing both reviewers comments.

I only have one remaining minor comment. I think that I now understand the logic the authors use now, but from the text it is still not clear that, as I understand it, 'SS' and 'DD' branch pairs do not perfectly map to what are traditionally referred to as orthologs or paralogs.

E.g., in the case where a duplication happens on an internal branch of the species tree and then the paralogs diversify by speciation, the paralogous copies within one species (that is, copy 1 and copy 2 in species A) will be considered and 'SS' branch pair even though we would normally think of these as paralogs, since the duplication event is on a deeper internal branch with subsequent speciation events.

A little bit of text and maybe a supplemental diagram to clarify what may otherwise end up a common misunderstanding would be a valuable minor addition prior to publication. It may also be useful to emphasize a little bit more in the text that (at least as I know understand it) this analysis focuses on "convergent gene duplication", that is AA changes that occur repeatedly after duplication events, not "convergence between gene duplicates" which implies (to me anyway) looking at convergence between paralogous copies within the same gene family.

Our ref: NATECOLEVOL-220316177A

1st September 2022

34Dear Dr. Fukushima,

Thank you for your patience as we've prepared the guidelines for final submission of your Nature Ecology & Evolution manuscript, "Detecting macroevolutionary genotype-phenotype associations using error-corrected rates of protein convergence" (NATECOLEVOL-220316177A). Please carefully follow the step-by-step instructions provided in the attached file, and add a response in each row of the table to indicate the changes that you have made. Please also check and comment on any additional marked-up edits we have proposed within the text. Ensuring that each point is addressed will help to ensure that your revised manuscript can be swiftly handed over to our production team.

****We would like to start working on your revised paper, with all of the requested files and forms, as soon as possible (preferably within two weeks). Please get in contact with us immediately if you anticipate it taking more than two weeks to submit these revised files.****

In recognition of the time and expertise our reviewers provide to Nature Ecology & Evolution's editorial process, we would like to formally acknowledge their contribution to the external peer review of your manuscript entitled "Detecting macroevolutionary genotype-phenotype associations using error-corrected rates of protein convergence". For those reviewers who give their assent, we will be publishing their names alongside the published article.

Nature Ecology & Evolution offers a Transparent Peer Review option for new original research manuscripts submitted after December 1st, 2019. As part of this initiative, we encourage our authors to support increased transparency into the peer review process by agreeing to have the reviewer comments, author rebuttal letters, and editorial decision letters published as a Supplementary item. When you submit your final files please clearly state in your cover letter whether or not you would like to participate in this initiative. Please note that failure to state your preference will result in delays in accepting your manuscript for publication.

Cover suggestions

As you prepare your final files we encourage you to consider whether you have any images or illustrations that may be appropriate for use on the cover of Nature Ecology & Evolution.

Covers should be both aesthetically appealing and scientifically relevant, and should be supplied at the best quality available. Due to the prominence of these images, we do not generally select images

35featuring faces, children, text, graphs, schematic drawings, or collages on our covers.

Nature Ecology & Evolution has now transitioned to a unified Rights Collection system which will allow our Author Services team to quickly and easily collect the rights and permissions required to publish your work. Approximately 10 days after your paper is formally accepted, you will receive an email in providing you with a link to complete the grant of rights. If your paper is eligible for Open Access, our Author Services team will also be in touch regarding any additional information that may be required to arrange payment for your article.

Please note that *Nature Ecology & Evolution* is a Transformative Journal (TJ). Authors may publish their research with us through the traditional subscription access route or make their paper immediately open access through payment of an article-processing charge (APC). Authors will not be required to make a final decision about access to their article until it has been accepted. [Find out more about Transformative Journals](https://www.springernature.com/gp/open-research/transformative-journals)

Authors may need to take specific actions to achieve [compliance with funder and institutional open access mandates](https://www.springernature.com/gp/open-research/funding/policy-compliance-faqs). If your research is supported by a funder that requires immediate open access (e.g. according to [Plan S principles](https://www.springernature.com/gp/open-research/plan-s-compliance)) then you should select the gold OA route, and we will direct you to the compliant route where possible. For authors selecting the subscription publication route, the journal's standard licensing terms will need to be accepted, including [self-archiving-and-license-to-publish](https://www.nature.com/nature-portfolio/editorial-policies/self-archiving-and-license-to-publish). Those licensing terms will supersede any other terms that the author or any third party may assert apply to any version of the manuscript.

[REDACTED]

[REDACTED]

Reviewer #1:

Remarks to the Author:

Re-review of "Detecting macroevolutionary genotype-phenotype associations using error-corrected rates of protein convergence"

The updated manuscript clarifies many of the issues raised in the first round of review, and I thank the authors for their many changes and updates. I still do not really agree with the focus on possible genotype-phenotype associations, but I at least agree that convergent evolution is one way to (indirectly) try to identify such associations.

Two remaining minor points that I don't feel are adequately addressed yet:

- "demonstrating that genotype-phenotype associations frequently occur on macroevolutionary scales" (lines 67-68). What does this mean? Are the authors trying to say something about the origins of traits, or our ability to detect them? What does convergent evolution have to do with the existence of such associations? Presumably there is an association between genotype and phenotype regardless of whether there is convergence.

- The authors have clarified that they have tested for convergence among duplicated genes within the same species. What phenotypes can duplicated genes from the same species be associated with? Related to this, the authors have also clarified that when one duplicate is lost, the remaining branch is "contaminated" with substitutions from the neighboring branch (because they cannot be separated). But what happens when a single-copy gene is lost from just one species? The previous answer implies there are none of these, but I can't tell whether this is the case or whether there was some confusion about what I meant by "lost genes". Can the authors please clarify?

Reviewer #2:

Remarks to the Author:

The authors have added a number of useful analyses, figures, and text that have in my opinion done an excellent job of addressing both reviewers comments.

37I only have one remaining minor comment. I think that I now understand the logic the authors use now, but from the text it is still not clear that, as I understand it, 'SS' and 'DD' branch pairs do not perfectly map to what are traditionally referred to as orthologs or paralogs.

E.g., in the case where a duplication happens on an internal branch of the species tree and then the paralogs diversify by speciation, the paralogous copies within one species (that is, copy 1 and copy 2 in species A) will be considered and 'SS' branch pair even though we would normally think of these as paralogs, since the duplication event is on a deeper internal branch with subsequent speciation events.

A little bit of text and maybe a supplemental diagram to clarify what may otherwise end up a common misunderstanding would be a valuable minor addition prior to publication. It may also be useful to emphasize a little bit more in the text that (at least as I know understand it) this analysis focuses on "convergent gene duplication", that is AA changes that occur repeatedly after duplication events, not "convergence between gene duplicates" which implies (to me anyway) looking at convergence between paralogous copies within the same gene family.

Author Rebuttal, first revision:

Response to reviewer comments

We thank the reviewers for their thorough and thoughtful reviews. All comments are addressed in a point-by-point fashion below. We believe this has resulted in an improved manuscript that is appropriate for a *Nature Ecology & Evolution* audience.

Reviewer #1

The updated manuscript clarifies many of the issues raised in the first round of review, and I thank the authors for their many changes and updates. I still do not really agree with the focus on possible genotype-phenotype associations, but I at least agree that convergent evolution is one way to (indirectly) try to identify such associations.

Two remaining minor points that I don't feel are adequately addressed yet:

Response: We thank the reviewer for their helpful comments and appreciations to our revision.

-"demonstrating that genotype-phenotype associations frequently occur on macroevolutionary scales" (lines 67-68). What does this mean? Are the authors trying to say something about the origins of traits, or our ability to detect them? What does convergent evolution have to do with the existence of such associations? Presumably there is an association between genotype and phenotype regardless of whether there is convergence.

Response: Convergence is not a strict requirement of genotype-phenotype associations, but please note that, without convergence (i.e., repeated evolution), it is extremely difficult to statistically link phenotypes and responsible genotypes because of the overwhelming numbers of unrelated mutations that accumulate along lineages. Here, this dependent clause is followed by a previous clause that discusses the frequency of convergence, and the complete sentence is shown below.

"A meta-analysis reported that 111 out of 1,008 loci had been convergently modified to attain common phenotypic innovations, sometimes even between different phyla⁴, demonstrating that genotype-phenotype associations frequently occur on macroevolutionary scales."

To make it clear that this sentence is talking about the observed frequency, we revised it as below.

Change: A meta-analysis reported that 111 out of 1,008 loci had been convergently modified to attain common phenotypic innovations, sometimes even between different phyla⁴, illustrating that genotype-phenotype associations are frequently observed on macroevolutionary scales.

-The authors have clarified that they have tested for convergence among duplicated genes within the same species. What phenotypes can duplicated genes from the same species be associated with?

Response: This concern should be a comment on the following description in our earlier response letter.

"In the genome-scale analysis, DD branch pairs may be connected to different species, or to the same species if successive duplications happen in the lineage."

Genotype-phenotype associations are one of the main subjects of this paper, but not all analyses involve them. The analysis explained here (Fig. 2c) was designed to evaluate the effect of gene duplication on the protein convergence rate but not genotype-phenotype associations.

Related to this, the authors have also clarified that when one duplicate is lost, the remaining branch is "contaminated" with substitutions from the neighboring branch (because they cannot be separated). But what happens when a single-copy gene is lost from just one species? The previous answer implies there are none of these, but I can't tell whether this is the case or whether there was some confusion about what I meant by "lost genes". Can the authors please clarify?

Response: Lost gene lineages are unanalyzable and therefore were not included in our analysis. In response to this comment, we clarified as follows.

Change: Our methods are compatible with gene losses, but lost gene lineages lead to the absence of ancestor-descendant branches that could otherwise be analyzed and contribute to the informativeness of the data.

Reviewer #2

The authors have added a number of useful analyses, figures, and text that have in my opinion done an excellent job of addressing both reviewers comments.

Response: We thank the reviewer for their helpful comments.

I only have one remaining minor comment. I think that I now understand the logic the authors use now, but from the text it is still not clear that, as I understand it, 'SS' and 'DD' branch pairs do not perfectly map to what are traditionally referred to as orthologs or paralogs.

E.g., in the case where a duplication happens on an internal branch of the species tree and then the paralogs diversify by speciation, the paralogous copies within one species that is, copy 1 and copy 2 in species A) will be considered and 'SS' branch pair even though we would normally think of these as paralogs, since the duplication event is on a deeper internal branch with subsequent speciation events.

A little bit of text and maybe a supplemental diagram to clarify what may otherwise end up a common misunderstanding would be a valuable minor addition prior to publication. It may also be useful to emphasize a little bit more in the text that (at least as I know understand it) this analysis focuses on "convergent gene duplication", that is AA changes that occur repeatedly after duplication events, not "convergence between gene duplicates" which implies (to me anyway) looking at convergence between paralogous copies within the same gene family.

Response: SS branches and paralogy are not mutually exclusive, as the reviewer pointed out, by definition. Because we did not examine the effects of orthology/paralogy, we carefully avoided these terminologies and consistently referred the branching events we tested, i.e., combinations of gene duplication (D) and speciation (S). A diagram of branching events is provided in Fig. 2c, and, in response to this comment, we revised the manuscript as follows.

Change: To distinguish these possibilities, we compared the convergence rates of branch pairs after two separate speciation (SS) events and branch pairs after two independent gene duplications (DD) (Fig. 2c; Supplementary Text 9; Supplementary Fig. 8d,e). It should be noted that this analysis does not compare orthologs versus paralogs but assesses the effect of gene duplication, relative to the baseline mode of protein sequence evolution after speciation. Strikingly, gene duplication significantly decreased convergence rates ($P \approx 0$, $W = 23.0$, as determined by a two-sided Brunner–Munzel test; Fig. 2c).

Final Decision Letter:

12th October 2022

Dear Dr Fukushima,

We are pleased to inform you that your Article entitled "Detecting macroevolutionary genotype-phenotype associations using error-corrected rates of protein convergence", has now been accepted for publication in Nature Ecology & Evolution.

Over the next few weeks, your paper will be copyedited to ensure that it conforms to Nature Ecology and Evolution style. Once your paper is typeset, you will receive an email with a link to choose the appropriate publishing options for your paper and our Author Services team will be in touch regarding any additional information that may be required

You will not receive your proofs until the publishing agreement has been received through our system

Due to the importance of these deadlines, we ask you please us know now whether you will be difficult to contact over the next month. If this is the case, we ask you provide us with the contact information (email, phone and fax) of someone who will be able to check the proofs on your behalf, and who will be available to address any last-minute problems . Once your paper has been scheduled for online publication, the Nature press office will be in touch to confirm the details.

Acceptance of your manuscript is conditional on all authors' agreement with our publication policies (see www.nature.com/authors/policies/index.html). In particular your manuscript must not be published elsewhere and there must be no announcement of the work to any media outlet until the publication date (the day on which it is uploaded onto our web site).

Please note that *Nature Ecology & Evolution* is a Transformative Journal (TJ). Authors may publish their research with us through the traditional subscription access route or make their paper immediately open access through payment of an article-processing charge (APC). Authors will not be required to make a final decision about access to their article until it has been accepted. [Find out more about Transformative Journals](https://www.springernature.com/gp/open-research/transformative-journals)

Authors may need to take specific actions to achieve [compliance with funder and institutional open access mandates](https://www.springernature.com/gp/open-research/funding/policy-compliance-faqs). If your research is supported by a funder that requires immediate open access (e.g. according to [a](https://www.springernature.com/gp/open-research/funding/policy-compliance-faqs)

42[Plan S principles](https://www.springernature.com/gp/open-research/plan-s-compliance)) then you should select the gold OA route, and we will direct you to the compliant route where possible. For authors selecting the subscription publication route, the journal's standard licensing terms will need to be accepted, including <https://www.nature.com/nature-portfolio/editorial-policies/self-archiving-and-license-to-publish>. Those licensing terms will supersede any other terms that the author or any third party may assert apply to any version of the manuscript.

We welcome the submission of potential cover material (including a short caption of around 40 words) related to your manuscript; suggestions should be sent to Nature Ecology & Evolution as electronic files (the image should be 300 dpi at 210 x 297 mm in either TIFF or JPEG format). Please note that such pictures should be selected more for their aesthetic appeal than for their scientific content, and that colour images work better than black and white or grayscale images. Please do not try to design a cover with the Nature Ecology & Evolution logo etc., and please do not submit composites of images related to your work. I am sure you will understand that we cannot make any promise as to whether any of your suggestions might be selected for the cover of the journal.

You can generate the link yourself when you receive your article DOI by entering it here: <http://authors.springernature.com/share>.

[REDACTED]

P.S. Click on the following link if you would like to recommend Nature Ecology & Evolution to your librarian <http://www.nature.com/subscriptions/recommend.html#forms>

** Visit the Springer Nature Editorial and Publishing website at http://editorial-jobs.springernature.com?utm_source=ejp_NEcoE_email&utm_medium=ejp_NEcoE_email&utm_campaign=ejp_NEcoE for more information about our career opportunities. If you have any questions please click [here](mailto:editorial.publishing.jobs@springernature.com).**